# The Out-of-sample Extensions of t-SNE:
# From Gradient Descent to Fixed-point Iteration Algorithms

**Paul Honeine**                                              *paul.honeine@univ-rouen.fr*
*Univ Rouen Normandie*
*INSA Rouen Normandie*
*Université Le Havre Normandie*
*Normandie Univ*
*LITIS UR 4108*
*F-76000 Rouen, France*

**Reviewed on OpenReview:** *https://openreview.net/forum?id=kYwq49F8Gt*

## Abstract

This paper addresses the out-of-sample extension of the t-distributed stochastic neighbor embedding (t-SNE), namely extending the embedding to other data that were not considered in the training of the t-SNE. We demonstrate the ease of deriving the out-of-sample extension of t-SNE, thanks to the proper nature of t-SNE, namely without any auxiliary model. Several resolution strategies are devised, from gradient descent to fixed-point iteration algorithms. Moreover, we establish several theoretical findings that allow to understand the underlying optimization mechanism of the fixed-point iteration, by providing several appealing properties, including connections with the mean shift algorithm and the resolution of the pre-image problem in Machine Learning. Experimental results on three well-known real data sets show the relevance and efficiency of the proposed out-of-sample methods, with the repulsion-free fixed-point iteration outperforming the other methods.

## 1 Introduction

Considered as one of the fundamental topics in data science, data visualization is of great importance for understanding and interpreting the structure of large volumes of data (Donoho, 2017). Among the different families of statistical Machine Learning (ML) methods developed for data visualization and dimensional reduction, the t-distributed stochastic neighbor embedding (t-SNE) method is arguably one of the most popular thanks to its outstanding performance on a wide range of applications (van der Maaten & Hinton, 2008; Ghojogh et al., 2023; Anvari et al., 2025). This method relies on a representation of the data in a low-dimensional space according to a Student t-distribution. The optimization problem is the minimization of the Kullback-Leibler (KL) divergence between the distributions in both spaces. However, due to the poor convergence of the gradient descent approach, several acceleration tricks are required to enhance the convergence, such as early exaggeration and a momentum term, as well as other optimization schemes such as Dijkstra's algorithm (Markov, 2025) and particle swarm optimization (Allaoui et al., 2025; Lu & Calder, 2025). Recently, t-SNE has attracted much research, with many variants proposed in the literature and some recent guarantees and theoretical foundations. See for instance Shaham & Steinerberger (2017); Linderman & Steinerberger (2019); Cai & Ma (2022); Murray & Pickarski (2024); Iwasaki & Hino (2025).

While several variants of t-SNE have been proposed in the literature, they rely on embedding exclusively the training data. However, they do not allow extending the embedding to other data not considered in the training phase. This so-called out-of-sample (OoS) extension of embedding methods is of great interest in manifold learning and data visualization. Principal Component Analysis (PCA) and its nonlinear variants have a straightforward mechanism for an OoS extension, with simple linear or nonlinear projections (Honeine, 2012). However, the PCA approach seems to be an exception in the landscape of manifold learning and

data visualization methods. Spectral clustering methods, local linear embedding (LLE), isomap, Laplacian eigenmap, and many other methods rely on an eigendecomposition of some similarity/neighborhood matrix, and therefore researchers have proposed OoS extensions based on a Nyštöm approximation (Bengio et al., 2003) or with a linear reconstruction and kernel mapping for LLE (Ghojogh et al., 2020). More recently, OoS extensions have been proposed for multidimensional scaling (Herath et al., 2021) and for graph embedding (adjacency and Laplacian spectral embeddings) (Levin et al., 2021). These approaches are often generic and independent of the dimensionality reduction method, relying on optimization (Bunte et al., 2012), kernel mapping (Gisbrecht et al., 2012) or an auxiliary neural network (Espadoto et al., 2023). See (Reichmann et al., 2024) for a recent survey.

There has been an increasing interest in an OoS extension of t-SNE. Since t-SNE and its variants are designed to preserve, as much as possible, the local neighborhood in both spaces, a simple strategy for OoS embedding is to use the $k$-nearest neighbors, as proposed by the median-based heuristic. The latter identifies for each OoS sample its $k$-nearest neighbors and positions its embedding in the low-dimensional space at the median t-SNE location of these $k$ references. This median-based heuristic was explored by Kobak & Berens (2019) to use t-SNE for single-cell transcriptomics, and was also used as a baseline in (Poličar et al., 2023) and (Aghasanli & Angelov, 2025). Candel & Naccache (2021) proposed to minimize two cost functions, one for the embedding shape and the other one for the support embedding's match. Other resolutions replace the classical t-SNE with a parametric embedding, in the same spirit as PCA's projection map, such as with parametric versions that approximate the t-SNE (Van Der Maaten, 2009; Damrich et al., 2023) or UMAP (Sainburg et al., 2021). Other strategies explore kernel-based variants of t-SNE, which enable the use of kernel mapping (Gisbrecht et al., 2015; Zhang et al., 2021). It is worth noting that dimensionality reduction and kernel-based models are interconnected through the OoS formalism of the pre-image problem in ML (Honeine & Richard, 2011). Nevertheless, all these developments redefine t-SNE through some approximations, thus modifying the embedding and requiring the development of appropriate optimization schemes.

To the best of our knowledge, there are only a couple of studies that have considered the OoS extension of t-SNE through an objective function similar to the one of t-SNE, namely in the same spirit as the KL divergence. However, this leads to a non-convex optimization problem, with major difficulties in devising relevant optimization schemes. For instance, Berman et al. (2014) proposed a local optimization with the Nelder-Mead Simplex algorithm, while Poličar et al. (2023) considered a similar strategy as the standard t-SNE with an omitted symmetrization in order to make the optimization tractable. Moreover, they enhanced the optimization by starting with the median-based heuristic. Such strategies inherit the same weaknesses as the training of the traditional t-SNE, thus suffering from initialization issues and requiring an adequate fine-tuning of both the learning rate and the momentum term (Poličar et al., 2024).

This paper studies the OoS extension of t-SNE, by providing methodological, algorithmic, and theoretical advances on this problem. We demonstrate that the underlying formulation of t-SNE has an intuitive OoS extension, without the need to use off-the-shelf techniques that modify the model (such as kernel mapping or parametric models). To this end, we follow the same derivations as those conducted for the t-SNE algorithm in order to derive its OoS extension. This yields a simple gradient descent scheme, in the same spirit as t-SNE and the extensions studied in (Berman et al., 2014; Poličar et al., 2023). While such a strategy suffers from the same weaknesses as the conventional algorithm of t-SNE, we devise fixed-point iteration algorithms that overcome such weaknesses, such as overcoming stepsize tuning issues. Of particular interest, we devise an efficient fixed-point iteration method through a repulsion-free optimization, with interesting properties. We establish major theoretical findings on the optimization mechanism of the proposed fixed-point iteration, by demonstrating that each iteration seeks the minimum of a quadratic surrogate function, and can be derived as Newton's method. Moreover, we provide connections with the celebrated mean-shift clustering method and the resolution of the pre-image problem in ML. Finally, extensive experimental results demonstrate the relevance of the proposed methods on several well-known real data sets: MNIST, Fashion MNIST, and the Mouse Retina data sets (Macosko et al., 2015).

The main contributions of this paper are as follows:

- We show that it is straightforward to derive an OoS extension of t-SNE, by following similar derivations as the ones given in the original t-SNE paper.

- We devise several algorithms for the OoS extension of t-SNE, with gradient descent —in the same spirit as the original t-SNE training algorithm— and fixed-point iteration schemes.

- We bring forth theoretical foundations that allow us to understand the underlying mechanism in the resolution of the OoS extension of t-SNE with the proposed fixed-point iteration. In particular, we prove that each iteration (*i*) seeks the minimum of a quadratic surrogate function, and (*ii*) corresponds to a step of Newton's method.

- We provide connections with the literature of (*i*) the gradient of density and its well-known mean shift algorithm for mode seeking, and (*ii*) the resolution of the pre-image problem in ML, thus allowing us to bridge the gap between these different frameworks.

- We provide experimental results on three real data sets that demonstrate the relevance of these methodological and theoretical findings.

The rest of the paper is organized as follows. Section 2 sets the stage with some background material on t-SNE, as well as its main enhancement techniques and related work. |Section 3 presents the proposed methods for OoS extension of t-SNE with its resolution using a gradient descent optimization scheme. Section 4 introduces a fixed-point iteration scheme for the optimization and reveals appealing properties. Section 5 provides several theoretical results that provide solid foundations for the algorithmic developments. Section 6 provides comprehensive experimental results on three real data sets, and Section 7 concludes this paper with a conclusion and future works.

## 2 Background Material on t-SNE and Related Work

### 2.1 Preliminaries on t-SNE

Given a set of $n$ samples $x_1, x_2, \ldots, x_n$ in a high-dimensional space $\mathbb{X} \subset \mathbb{R}^D$, t-SNE seeks to embed them in a lower-dimensional space $\mathbb{Y} \subset \mathbb{R}^d$, yielding $y_1, y_2, \ldots, y_n$, by preserving the affinities between the samples in both spaces. t-SNE measures such affinity between two samples by examining their probability of being neighbors. In the original space $\mathbb{X}$, such probability for $x_i$ being the neighbor of $x_j$ takes the form of a Gaussian distribution

$$p_{i|j} = \frac{\exp(-\|x_i - x_j\|^2/2\sigma_j^2)}{\sum_{k \neq i} \exp(-\|x_i - x_k\|^2/2\sigma_j^2)}, \tag{1}$$

where $p_{i|i} = p_{j|j} = 0$ and where the bandwidth parameter $\sigma_j$ is obtained using a given perplexity value[1]. In order to consider a symmetric criterion, t-SNE uses the joint probabilities:

$$p_{i,j} = \frac{p_{i|j} + p_{j|i}}{2n}. \tag{2}$$

Similarly, the affinity in the embedded space is measured using a probability distribution. In order to overcome the *crowding problem* and compensate for mismatched dimensionalities, the Student t-distribution with one degree of freedom (namely, the Cauchy distribution) is introduced by van der Maaten & Hinton (2008), namely for any $y_i$ and $y_j$

$$q_{i,j} = \frac{(1 + \| y_i - y_j \|^2)^{-1}}{\sum_{k \neq l}(1 + \| y_k - y_l \|^2)^{-1}}, \tag{3}$$

---

[1]The perplexity corresponds to the conditional distribution of all the other samples given a sample. For each sample $x_i$, the value of the bandwidth parameter $\sigma_i$ is obtained such that the perplexity is $2^{-\sum_j p_{j|i} \log_2 p_{j|i}}$.

with similarly $q_{i,i} = q_{j,j} = 0$. The objective function is the KL divergence between the joint distributions in the original space and the corresponding ones in the embedded space, namely

$$\Xi(P||Q) = \sum_{\substack{i,j \in \{1,\ldots,n\} \\ i \neq j}} p_{i,j} \log \frac{p_{i,j}}{q_{i,j}}. \tag{4}$$

The t-SNE algorithm minimizes this objective function using a gradient descent scheme. The gradient of this cost function at any $y_i$ boils down to

$$\nabla_{y_i} \Xi(y_i) = 4 \sum_j (p_{i,j} - q_{i,j})(y_i - y_j)(1 + \| y_i - y_j \|^2)^{-1}. \tag{5}$$

The gradient descent takes the form of the iterative rule

$$y_i^{t+1} = y_i^t - \eta^t \nabla_{y_i^t} \Xi(y_i^t), \tag{6}$$

for a positive stepsize parameter $\eta^t$ (also called learning rate).

In the original paper of t-SNE, as well as in most papers of the literature, the minimization of this cost function is performed using a gradient descent with some techniques to improve the convergence. These acceleration techniques are a momentum term and an early exaggeration stage, both introduced first for t-SNE by van der Maaten & Hinton (2008):

- Early Exaggeration: An exaggeration corresponds to multiplying the probabilities in the original space with a coefficient $\alpha > 1$. An early exaggeration is recommended to enhance convergence in the early iterations of the gradient descent (e.g., during the first 250 iterations). Late exaggeration, namely operating exaggeration in the end of the optimization rather than in the beginning, forces the clusters to be tighter and further apart, and therefore enhances the visualizations, as advocated by Linderman et al. (2017).

- Momentum: A momentum term (also related to a delta-bar-delta scheme) can be considered in the update rule, taking the form

$$y_i^{t+1} = y_i^t - \eta^t \nabla_{y_i^t} \Xi(y_i^t) + \eta_{\text{mom}}^t \left( y_i^t - y_i^{t-1} \right), \tag{7}$$

where $\eta_{\text{mom}}^t$ represents the momentum at iteration $t$ with a value that needs to be fine-tuned.

## 2.2 On the Out-of-sample Extension of t-SNE

The t-SNE algorithm embeds the training samples $x_1, x_2, \ldots, x_n \in \mathbb{X}$ into $y_1, y_2, \ldots, y_n \in \mathbb{Y}$ in a low dimensional space. However, it does not define the embedding function. Therefore, once we have the samples in both representations, it does not allow us to project any other sample, in contrast to PCA where one can easily define the projection matrix. In the following, we study the embedding of a new sample $x_\ell \in \mathbb{X}$ to a reference background t-SNE, namely, an already available t-SNE map generated by training samples Therefore, our aim is to compute its corresponding $y_\ell$ in the low-dimensional space $\mathbb{Y}$, hence representing it within the embedded $y_1, y_2, \ldots, y_n \in \mathbb{Y}$.

To the best of our knowledge, there are only two studies that have considered the OoS extension of t-SNE through an objective function similar to that of t-SNE, namely, (5). The main idea is to enforce the probabilities in the two spaces to be as similar as possible, by minimizing the KL divergence between the distributions

$$p_{i|\ell} = \frac{\exp(-d(x_\ell, x_i)^2 / 2\sigma_\ell^2)}{\sum_k \exp(-d(x_\ell, x_k)^2 / 2\sigma_\ell^2)}, \tag{8}$$

and

$$q_{i|\ell} = \frac{(1 + \| y_\ell - y_i \|^2)^{-1}}{\sum_k (1 + \| y_\ell - y_k \|^2)^{-1}}. \tag{9}$$

In (Berman et al., 2014), $d(\cdot, \cdot)$ in (8) are computed using the KL divergence between the training samples and the OoS samples. However, this modification led to a more complex non-convex objective function, with the resulting problem being difficult to optimize. To alleviate these difficulties, the authors explored a local optimization using the Nelder-Mead Simplex algorithm on a weighted average of samples.

More recently, Poličar et al. (2023) investigated (8)-(9) in the same spirit as the standard t-SNE. To make the optimization problem tractable, they omitted the symmetrization step (2) as in conventional SNE (Hinton & Roweis, 2002), thus solving a modified loss function where $p_{i,j}$ and $q_{i,j}$ in (4) were substituted with $p_{i|\ell}$ and $q_{i|\ell}$. The resulting gradient of the modified loss at any $y_\ell$ becomes[2]

$$\nabla_{y_\ell} \Xi(y_\ell) = 2 \sum_j (p_{j|\ell} - q_{j|\ell})(y_\ell - y_j)(1 + \| y_\ell - y_j \|^2)^{-1}. \tag{10}$$

In this case, each OoS $y_\ell$ is processed independently from other OoS samples, while the embeddings of the training data remain unchanged. Although using a non-symmetrized version, a gradient descent using (10) inherits the same weaknesses as standard t-SNE, such as requiring an adapted learning rate scheme with a fine-tuned momentum. The experiments conducted by Poličar et al. (2023) were carried out with a median-based heuristic initialization and a very small learning rate of $\eta = 0.1$, thus barely exploring beyond the initialization, which is different from conventional implementations of t-SNE with a learning rate of $\eta = 200$ by default (Poličar et al., 2024).

## 3 OoS Algorithms for t-SNE

The literature review shows that there is no work that considers the exact t-SNE optimization problem and resolution techniques to solve the OoS extension. As outlined in the previous section, they either explore a hybrid resolution, such as a non-symmetrized variant as in the traditional SNE, or surrogate resolutions, such as the median-based heuristic or the Nelder-Mead Simplex algorithm. In this section, we show that one can easily derive an OoS extension of t-SNE, by fully leveraging the t-SNE mechanisms. In light of the derivations carried out in the previous section for the classical t-SNE, we first derive a gradient descent scheme that embeds new samples into a t-SNE reference map already existing. Then, we establish a fixed-point iteration algorithm. For the sake of notation simplicity, we consider only a single OoS sample to be embedded in a t-SNE map. See Section 3.3 for extensions to multiple OoS samples.

### 3.1 The Gradient Descent Scheme

Let $x_\ell \in \mathbb{X}$ be the OoS and $y_\ell$ its embedding to be estimated in the low-dimensional space $\mathbb{Y}$ based on a reference t-SNE background, which is the $n$ samples $x_1, x_2, \ldots, x_n \in \mathbb{X}$ and their embedding $y_1, y_2, \ldots, y_n \in \mathbb{Y}$.

For this purpose, we consider the affinities in both spaces, where the affinities are computed over the augmented data set, namely the $n$ samples of the training data and the OoS $y_\ell$. In this case, the affinity measure in $\mathbb{X}$ is

$$p_{i,j} = \frac{p_{i|j} + p_{j|i}}{2(n + 1)}, \tag{11}$$

and in $\mathbb{Y}$ is $q_{i,j}$, with $p_{i|j}$ and $q_{i,j}$ of the same form as in (1) and (3), respectively, where the denominators take into account the OoS in the augmented data set. The corresponding KL divergence on the augmented data set is

$$\Xi_{\text{aug}} = \sum_{i,j \in \{1,\ldots,n\} \cup \{\ell\}} p_{i,j} \log \frac{p_{i,j}}{q_{i,j}}. \tag{12}$$

---

[2]In (Poličar et al., 2024), there is a sign error in the expression of the gradient for embedding new samples. This error is corrected in this present paper.

By following the same procedure as in the standard t-SNE, we get the following gradient of this objective function, computed here at the OoS embedding $y_\ell$:

$$\nabla_{y_\ell}\Xi_{\text{aug}}(y_\ell) = 4\sum_{j=1}^{n}(p_{\ell,j} - q_{\ell,j})w_{\ell,j}(y_\ell - y_j), \tag{13}$$

where

$$w_{\ell,j} = (1 + \|y_\ell - y_j\|^2)^{-1}. \tag{14}$$

The gradient descent takes the form of the iterative rule

$$y_\ell^{t+1} = y_\ell^t - \eta^t\nabla_{y_\ell^t}\Xi_{\text{aug}}(y_\ell^t), \tag{15}$$

for a positive learning rate parameter $\eta^t$. This yields the gradient descent algorithm for OoS extension of t-SNE, with expressions having similar forms to the one used for t-SNE (5)-(6).

We can rewrite the gradient as the difference between two terms

$$\nabla_{y_\ell}\Xi_{\text{aug}}(y_\ell) = 4\Big(\underbrace{\sum_{j=1}^{n}p_{\ell,j}w_{\ell,j}(y_\ell - y_j)}_{\text{Attraction term}} - \underbrace{\sum_{j=1}^{n}q_{\ell,j}w_{\ell,j}(y_\ell - y_j)}_{\text{Repulsion term}}\Big), \tag{16}$$

where the first term operates as an attraction term that pulls the OoS estimate $y_\ell$ towards similar training samples, while the second term operates as a repulsion term that attempts to separate $y_\ell$ from nearby training samples. The above rewriting shows that the resolution of an OoS through a gradient descent scheme has the same mechanism as t-SNE. As demonstrated by van der Maaten & Hinton (2008), such a gradient strongly repels dissimilar samples modeled by small pairwise distances thanks to the usage of the Student t-distribution $q$ defined in (3), in contrast to the conventional SNE.

The gradient descent scheme for OoS is of the same form as that of t-SNE (5)-(6). Therefore, it inherits the same weaknesses and convergence issues. We can borrow acceleration techniques introduced for the standard t-SNE to enhance the convergence of the proposed gradient descent scheme for OoS. In the same spirit as the momentum term included in (7), we can add momentum to the gradient descent (15). Early exaggeration (and possibly late exaggeration) can be easily injected into the proposed update rule, by substituting $p_{\ell,j}$ with $\alpha p_{\ell,j}$ for all $j = 1, 2, \ldots, n$ in the gradient (13).

Nevertheless, the resolution of the OoS with a gradient descent scheme remains challenging and cumbersome due to the necessity of fine-tuning all its parameters: the learning rate parameter $\eta^t$, the early exaggeration value $\alpha$ and its duration, and the momentum $\eta^t_{\text{mom}}$. These weaknesses are overcome by the fixed-point iteration algorithms introduced next.

## 3.2 Fixed-point Iteration Algorithms

We propose in this section a fixed-point iteration scheme for the OoS extension of t-SNE. To design a fixed-point iteration, one needs to write an expression of the form $y = f(y)$ at the optimum, namely at gradient nullification $\nabla_y\Xi(y) = 0$. This would allow one to define an iterative update rule as $y^{t+1} = f(y^t)$. There are infinitely many candidates, by considering for example $f(y) = y - \gamma\nabla_y\Xi(y)$ for any fixed $\gamma$ or by splitting the gradient as $\nabla_y\Xi(y) = \frac{1}{\gamma}(y - f(y))$. Since different splits can be carried out, the function $f(\cdot)$ is not unique, leading to different fixed-point iteration algorithms. In the following, we propose a simple strategy to define our fixed-point iteration.

To derive the fixed-point iteration, we nullify the gradient (13) at the optimum, which leads to the equation

$$y_\ell\sum_{j=1}^{n}w_{\ell,j}(p_{\ell,j} - q_{\ell,j}) = \sum_{j=1}^{n}w_{\ell,j}(p_{\ell,j} - q_{\ell,j})\,y_j.$$

From this expression, we derive the fixed-point iteration

$$y_\ell^{t+1} = \frac{\sum_j w_{\ell,j}^t (p_{\ell,j} - q_{\ell,j}^t) \, y_j}{\sum_j w_{\ell,j}^t (p_{\ell,j} - q_{\ell,j}^t)}, \tag{17}$$

where $w_{\ell,j}^t$ and $q_{\ell,j}^t$ are evaluated on $y_\ell^t$ and $y_j$. This fixed-point iteration is of the form $y_\ell^{t+1} = \sum_{j=1}^n \beta_j^t \, y_j$, where $\sum_{j=1}^n \beta_j^t = 1$ and $-1 \le \beta_j^t \le 1$, corresponding to an affine combination of the reference points $y_1, y_2, \ldots, y_n$. Moreover, by exploring different variants of the objective function, one could derive other fixed-point iteration algorithms. For instance, considering the non-symmetrization as (Poličar et al., 2023), the resulting gradient (10) allows us to provide the fixed-point iteration

$$y_\ell^{t+1} = \frac{\sum_j w_{\ell,j}^t (p_{j|\ell} - q_{j|\ell}^t) \, y_j}{\sum_j w_{\ell,j}^t (p_{j|\ell} - q_{j|\ell}^t)}. \tag{18}$$

Since the construction of the fixed-point iteration $f(\cdot)$ is not unique, we would like to point out that different formulations are possible other than (17), leading to different fixed-point iteration algorithms. For example, the split

$$y_\ell \sum_{j=1}^n w_{\ell,j} p_{\ell,j} = y_\ell \sum_{j=1}^n w_{\ell,j} q_{\ell,j} + \sum_{j=1}^n w_{\ell,j} (p_{\ell,j} - q_{\ell,j}) \, y_j \tag{19}$$

leads to the following fixed-point iteration

$$y_\ell^{t+1} = \frac{y_\ell^{t+1} \sum_j w_{\ell,j}^t q_{\ell,j}^t + \sum_j w_{\ell,j}^t (p_{\ell,j} - q_{\ell,j}^t) \, y_j}{\sum_j w_{\ell,j}^t p_{\ell,j}}. \tag{20}$$

Such a split was introduced by Yang et al. (2009) for the so-called HSSNE updating rule as a substitute for the gradient descent scheme of t-SNE. We could also explore other optimization schemes proposed initially for t-SNE, such as the Laplacian-inspired update rule proposed by van der Maaten (2010) or the nonnegative-based split of Yang et al. (2010) in the same spirit as the nonnegative matrix factorization. Our choice for the split that gives (17) is motivated by many fundamental properties, as detailed in the following sections.

For all these formulations, there is no guarantee that the fixed-point iteration is a contracting mapping. Moreover, experimental results demonstrate convergence issues, probably due to the affine convexity property, which is a by-product of the repulsion-attraction terms in the gradient (16). In practice, one could alleviate such convergence issues by considering high exaggeration values for (20), leading to a repulsion-free behavior as described in Section 4.

### 3.3 Addressing Multiple OoS Samples

In the following, we study the embedding of multiple OoS samples on the same t-SNE reference. We propose two different strategies, whether each OoS sample is estimated independently or dependently on the other OoS samples: the first one rewrites through linear algebra the proposed fixed-point iteration algorithms (17) for each OoS independently, while the second one revisits the KL divergence (12) by augmenting it to include all the OoS samples.

### 3.3.1 Independent Optimization Strategy

Let $\boldsymbol{Y}$ be the $d$-by-$n$ matrix of the estimated coordinates of $y_1, y_2, \ldots, y_n$, namely the t-SNE embedded samples of the $n$ training data $x_1, x_2, \ldots, x_n$. Let $m$ be the number of OoS samples. We address the estimate of each OoS independently of the other estimates, by estimating $m$ independently fixed-point iteration based on the $n$ reference t-SNE samples. The update rule (17) can be written in matrix form for all $m$ estimates as

$$y_\ell^{t+1} = \boldsymbol{Y} \left( \tfrac{1}{(\boldsymbol{p}_\ell - \boldsymbol{q}_\ell^t)^\top \boldsymbol{w}_\ell^t} (\boldsymbol{p}_\ell - \boldsymbol{q}_\ell^t) \odot \boldsymbol{w}_\ell^t \right), \tag{21}$$

where $\boldsymbol{p}_\ell$ and $\boldsymbol{q}_\ell^t$ are the column vectors of entries $p_{\ell,1}, p_{\ell,2}, \ldots, p_{\ell,n}$, and $q_{\ell,1}^t, q_{\ell,2}^t, \ldots, q_{\ell,n}^t$, respectively, and $\boldsymbol{w}_\ell^t$ is a column vector of entries $w_{\ell,1}^t, w_{\ell,2}^t, \ldots, w_{\ell,n}^t$. The operator $\odot$ denotes the Hadamard product (i.e.

element-wise multiplication). This fixed-point iteration can be easily extended to operate on the embedding of multiple OoS samples. For $m$ samples to be embedded in an existing t-SNE obtained from the $n$ training samples, the corresponding iteration takes the form

$$\boldsymbol{Y}_{\text{oos}}^{t+1} = \boldsymbol{Y}\big(((\boldsymbol{P} - \boldsymbol{Q}^t) \odot \boldsymbol{W}^t)\text{diag}((\boldsymbol{P} - \boldsymbol{Q}^t)^\top \boldsymbol{W}^t)^{-1}\big), \tag{22}$$

where $\boldsymbol{Y}_{\text{oos}}^{t+1}$ is the matrix of the $m$ estimated guesses at iteration $t + 1$, $\boldsymbol{P}$, $\boldsymbol{Q}^t$ and $\boldsymbol{W}^t$ are the matrices containing the column vectors $\boldsymbol{p}_\ell$, $\boldsymbol{q}_\ell^t$ and $\boldsymbol{w}_\ell^t$ for all the $m$ samples to be embedded. The operator $\text{diag}(\cdot)$ outputs a diagonal matrix by extracting the diagonal of its input matrix.

### 3.3.2 Joint Optimization Strategy

The second resolution for embedding multiple OoS samples is through the KL divergence, by augmenting it to include all the OoS samples. In the following, we denote by $x_{n+1}, x_{n+2}, \ldots, x_{n+m}$ the $m$ OoS samples, and their corresponding estimated $y_{n+1}^{t+1}, y_{n+2}^{t+1}, \ldots, y_{n+3}^{t+1}$. Then, the augmented KL divergence (12) becomes

$$\Xi_{\text{aug}} = \sum_{i,j \in \{1,\ldots,n\} \cup \{n+1,\ldots,n+m\}} p_{i,j} \log \frac{p_{i,j}}{q_{i,j}}. \tag{23}$$

The minimization of this objective function can be done with a gradient descent or a fixed-point iteration, as described respectively by (15) and (17). The only difference is that, for a given OoS $x_\ell$, the summations are now on all the other samples, namely the $n$ training data and the other $m-1$ OoS samples. The resulting resolutions lead to expressions similar to (22) for instance.

The choice of the resolution strategy depends on the problem at hand. The independent optimization carried out in the first strategy could be preferred in general, and more specifically when the number $m$ of OoS samples is large.

The joint optimization conducted in the second strategy can be promoted when we know that there is some relation between the $m$ of OoS samples, for instance. However, since the estimation of each $y_\ell^{t+1}$ depends on all the OoS estimates, it is expected that the second strategy has weaker convergence. However, experimental results show that both strategies have a small effect on the minimization of the KL divergence and on the Trustworthiness and Continuity metrics. See Section 5.4.2 for details.

### 3.4 Computational Complexity

The proposed fixed-point iteration has a low computational complexity and memory usage. For one update of a single OoS, the terms in brackets in (21) require $\mathcal{O}(2n)$ multiplications and $\mathcal{O}(n)$ divisions (for computing $\boldsymbol{w}_\ell^t$ at each iteration), while $\boldsymbol{p}_\ell$ is computed once and remains unchanged at each update. The matrix multiplication with $\boldsymbol{Y}$ is in $\mathcal{O}(2n)$.

When dealing with multiple OoS samples (22), the matrix multiplication with $\boldsymbol{Y}$ becomes in $\mathcal{O}(2nm)$ for embedding $m$ OoS samples. The operation $(\boldsymbol{P} - \boldsymbol{Q}^t) \odot \boldsymbol{W}^t$ requires $\mathcal{O}(nm)$ multiplications. Despite the most computationally expensive operation being $\boldsymbol{P}^\top \boldsymbol{W}^t$ with $\mathcal{O}(nm^2)$, it is not required in practice as we only need the diagonal elements of this multiplication, which can be done in $\mathcal{O}(nm)$ only.

Therefore, the computational complexity of the proposed algorithm is only linear in $n$, which makes it possible to operate on big data. The proposed algorithm can be implemented independently of the t-SNE algorithm, using either the conventional t-SNE algorithm (van der Maaten & Hinton, 2008) or the accelerated nearest-neighbors or the tree-based algorithms (van der Maaten, 2014).

### 3.5 On Gradient Descent versus Fixed-point Iteration Algorithms

Gradient descent schemes are explicitly aimed at optimization, namely minimizing a cost function. Fixed-point iteration is primarily for solving equations of the form $y = f(y)$, rather than minimizing a cost function. Thus, deriving a fixed-point iteration is generally more difficult.

Although the original t-SNE was introduced using a gradient descent scheme (van der Maaten & Hinton, 2008), several variants have been proposed to explore fixed-point iteration for optimization by following

different reformulations (Yang et al., 2009; van der Maaten, 2010; Yang et al., 2010). However, all these methods suffer from several drawbacks, mainly convergence issues, which make them less practical. On the other hand, gradient descent schemes are also not infallible, but they are often easier to use, with somewhat predictable convergence behavior under suitable learning rates.

These weaknesses of the fixed-point iteration extend naturally to the OoS extensions of t-SNE, as these drawbacks are inherit to the fixed-point iteration framework. Indeed, the construction of the fixed-point iteration function $f(\cdot)$ is not unique, and its efficiency heavily depends on its expression, namely on how the problem is reformulated as a fixed-point iteration. To ensure convergence, the function should satisfy certain conditions, such as being a contraction mapping. However, the aforementioned methods do not satisfy such these conditions.

In the following, we propose a repulsion-free fixed-point iteration method that does not suffer from these drawbacks, as demonstrated through theoretical foundations and experimental results.

## 4 Repulsion-free OoS Algorithms for t-SNE

There are multiple reasons for not considering the repulsion term. From a conceptual viewpoint, when one is seeking to position an OoS sample on a predefined t-SNE map with training samples as references, the major objective is the attraction term that pulls the OoS to the appropriate neighborhood. There is no particular reason for a repulsion term to separate it from its neighboring samples. Moreover, the study of the repulsion-free strategy allows us to address the high value of an exaggeration regime, either in early or late phases. It is well-known that exaggeration provides some interesting properties, such as clustering and several interesting theoretical foundations, as studied in (Linderman & Steinerberger, 2019; Cai & Ma, 2022). Finally, Section 4.2 provides solid theoretical findings that demonstrate the relevance of our strategy and the resulting fixed-point iteration scheme, by establishing that this fixed-point iteration can be derived directly from Newton's method.

The repulsion-free gradient descent algorithm is defined by the update rule (15), where the repulsion term of the gradient in (16) is omitted, namely

$$y_\ell^{t+1} = y_\ell^t - 4\eta^t \sum_{j=1}^n p_{\ell,j} w_{\ell,j}^t (y_\ell^t - y_j) \tag{24}$$

In the following, we study in detail the repulsion-free variant of the fixed-point iteration.

### 4.1 Repulsion-free Fixed-point Iteration

By considering the gradient (16) without the repulsion term, the fixed-point iteration (17) boils down to

$$y_\ell^{t+1} = \frac{\sum_j w_{\ell,j}^t p_{\ell,j} y_j}{\sum_j w_{\ell,j}^t p_{\ell,j}}, \tag{25}$$

which is also the repulsion-free variant of (20) based on the HSSNE split. The resulting pseudocode of the algorithm is given in Algorithm 1, illustrating the simplicity of its implementation. In the following, we provide some interesting properties of this fixed-point iteration, before establishing several theoretical findings.

This fixed-point iteration does not suffer from any parameter tuning, as opposed to gradient descent with its learning rate and momentum acceleration parameters. Moreover, the proposed update rule can be viewed as a gradient descent update (15) with a stepsize adapted at each iteration. We can easily see this by considering the gradient descent update with the stepsize parameter set to

$$\eta_*^t = \frac{1}{4\sum_j w_{\ell,j}^t p_{\ell,j}}, \tag{26}$$

---

**Algorithm 1:** The repulsion-free fixed-point iteration algorithm.
See Section 3.3 for the extension to multiple OoS samples.

---

**Data:**
$\{(x_1, y_1), \ldots, (x_n, y_n)\}$ (t-SNE training samples and their embeddings)
$x_\ell$ (the OoS)
$T$ (number of iterations)
**Result:**
$y_\ell$ (estimated embedding of $x_\ell$)

**1 begin**
**2**     Initialize $y_\ell^0$
**3**     Compute $p_{\ell,j}$ from (11) for all $(x_\ell, x_j)$, $j = 1, 2, \ldots n$
**4**     **for** $t = \ell$ **to** $T - 1$ **do**
**5**        Compute $w_{\ell,j}^t$ from (14) for all $(y_\ell^t, x_j)$, $j = 1, 2 \ldots n$
**6**        Apply the repulsion-free fixed-point iteration (25)
**7**     **end**
**8**     **Output:** $y_\ell^T$
**9 end**

---

which is nonnegative since $w_{\ell,j}^t, p_{\ell,j} > 0$. By injecting this stepsize in (15), we get the fixed-point iteration (25). It turns out that this adaptive stepsize allows optimized updates, as demonstrated by the theoretical findings given in Section 4.2, such as showing that each iteration corresponds to a Newton's step.

Moreover, the fixed-point iteration (25) is of the form

$$y_\ell^{t+1} = \sum_{j=1}^n \beta_j^t \, y_j, \tag{27}$$

where

$$\beta_j^t = \frac{w_{\ell,j}^t p_{\ell,j}}{\sum_j w_{\ell,j}^t p_{\ell,j}}. \tag{28}$$

Since $\sum_{j=1}^n \beta_j^t = 1$ and $\beta_j^t > 0$, we have a convex combination of $y_1, y_2, \ldots, y_n$. Therefore, each update $y_\ell^{t+1}$ belongs to the convex hull of the t-SNE embedded $y_1, y_2, \ldots, y_n$. This is a very interesting property, since it constrains the solution to not "overshoot" beyond the reference training data. Moreover, this illustrates the clustering behavior of the algorithm, which will be studied in Section 4.3.1 by connecting it to the mean shift algorithm.

With all these interesting properties, it is also worth noting that the exaggeration has no effect on the update rule (15), due to the normalization in its expression. Furthermore, the proposed method also does not require any particular initialization (see Section 5.4.2 for a study of the initialization sensitivity). This is not the case for t-SNE, as well as other related methods such as UMAP, since their optimizations rely heavily on the initialization, as corroborated by Kobak & Linderman (2019); Yang et al. (2021); Kobak & Linderman (2021).

In the following, we provide some theoretical results in favor of the repulsion-free fixed-point iteration for the OoS extension of t-SNE. Moreover, we provide connections to other methods, bridging the gap with the mean shift algorithm for mode seeking and the resolution of the pre-image problem in ML.

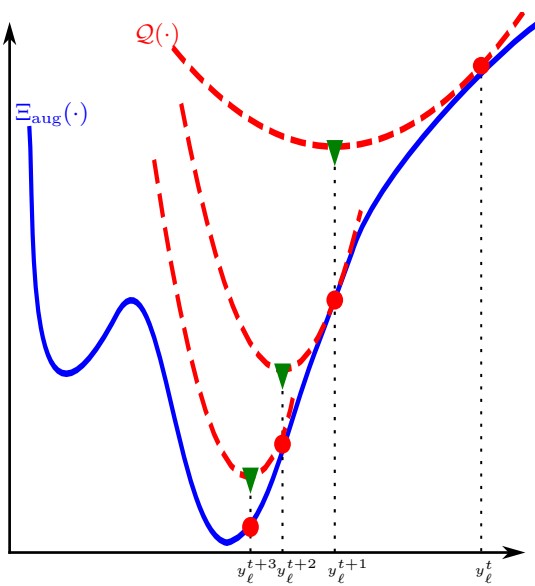

Figure 1: Illustration in 1D of the repulsion-free fixed-point iteration as a quadratic function minimization. To minimize the objective function $\Xi_{\text{aug}}(\cdot)$ (blue solid line), the repulsion-free fixed-point iteration updates a guess solution $y_\ell^t$ (red ●) to get $y_\ell^{t+1}$, which is the minimum (▼) of the quadratic function $\mathcal{Q}(\cdot)$ (red dashed parabola) that is tangent to $\Xi_{\text{aug}}(\cdot)$ at $y_\ell^t$. See Theorems 1 and 2 for more details.

## 4.2 Theoretical Insights on the Repulsion-free Fixed-point Iteration

Before proceeding, we derive the Hessian of the objective function under study, namely the Jacobian matrix of its gradient. The Hessian matrix at $y_\ell^t$ of $\Xi_{\text{aug}}(\cdot)$ defined in (12) is

$$\nabla_{y_\ell^t}^2 \Xi_{\text{aug}}(y_\ell^t) = 4 \sum_{j=1}^n p_{\ell,j}\Big((1 + \|y_\ell^t - y_j\|^2)^{-1}\boldsymbol{I} + (1 + \|y_\ell^t - y_j\|^2)^{-2}(y_\ell^t - y_j)(y_\ell^t - y_j)^\top\Big). \tag{29}$$

With $w_{\ell,j} = (1 + d_{\ell,j}^2)^{-1}$, we get

$$\nabla_{y_\ell^t}^2 \Xi_{\text{aug}}(y_\ell^t) = 4 \sum_{j=1}^n p_{\ell,j} w_{\ell,j}\Big(\boldsymbol{I} + w_{\ell,j}(y_\ell^t - y_j)(y_\ell^t - y_j)^\top\Big). \tag{30}$$

Therefore, the Hessian matrix is positive definite since it is a linear combination of positive definite matrices (identity and rank-one matrices) with positive coefficients.

The following theorem examines the optimization mechanism operated by the repulsion-free fixed-point iteration. See Figure 1 for an illustration.

**Theorem 1.** *The repulsion-free fixed-point iteration* (25) *at a guess* $y_\ell^t$ *seeks the minimization of the quadratic function*

$$\mathcal{Q}(y) = 2 \sum_{j=1}^n p_{\ell,j} w_{\ell,j}^t \|y - y_j\|^2 - c(y_\ell^t), \tag{31}$$

*with* $c(y_\ell^t) = -2 \sum_{j=1}^n p_{\ell,j}\Big(\log \frac{p_{\ell,j}}{q_{\ell,j}^t} - w_{\ell,j}^t \|y_\ell^t - y_j\|^2\Big)$ *being independent of* $y$. *Moreover, the quadratic function* $\mathcal{Q}(\cdot)$ *is tangent to* $\Xi_{aug}(\cdot)$ *at* $y_\ell^t$.

*Proof.* First, it is easy to see that $\Xi_{\mathrm{aug}}(y_\ell^t) = \mathcal{Q}(y_\ell^t)$. Second, the gradient of $\mathcal{Q}(y)$ with respect to its argument is

$$\nabla_y \mathcal{Q}(y) = 4 \sum_{j=1}^n p_{\ell,j} w_{\ell,j}^t (y - y_j), \tag{32}$$

thus its evaluation at $y_\ell^t$ is exactly the gradient $\nabla_y \Xi_{\mathrm{aug}}(y_\ell^t)$ given in (13). Therefore, $\mathcal{Q}(\cdot)$ is tangent to $\Xi_{\mathrm{aug}}(\cdot)$ at $y_\ell^t$. Moreover, $\mathcal{Q}(\cdot)$ has a minimum since its Hessian is positive definite. Finally, the optimum of this quadratic function is obtained when its gradient (32) is nullified, which yields the repulsion-free fixed-point iteration at $y_\ell^t$. □

This theorem allows us to understand how the repulsion-free fixed-point iteration works at each step. Figure 1 illustrates its operational functioning, where each iteration corresponds to the minimization of the quadratic function $\mathcal{Q}(\cdot)$. Thus, it can be viewed as a gradient descent with an adapted stepsize at each iteration.

The following theorem goes further by demonstrating that this stepsize is optimized using the curvature information as Newton's method.

**Theorem 2.** *The repulsion-free fixed-point iteration* (25) *at each guess $y_\ell^t$ operates as a step of Newton's method for the function $\mathcal{Q}(\cdot)$ defined in Theorem 1.*

*Proof.* The minimum of the quadratic function $\mathcal{Q}(\cdot)$ can be obtained from a single step of Newton's method, namely

$$y_\ell^{t+1} = y_\ell^t - \left( \nabla_{y_\ell^t}^2 \mathcal{Q}(y_\ell^t) \right)^{-1} \nabla_{y_\ell^t} \mathcal{Q}(y_\ell^t). \tag{33}$$

The Hessian of the quadratic function $\mathcal{Q}(\cdot)$ with respect to its argument is

$$\nabla_y^2 \mathcal{Q}(y) = 4 \sum_{j=1}^n p_{\ell,j} w_{\ell,j}^t \boldsymbol{I}. \tag{34}$$

Since $\left( \nabla_{y_\ell^t}^2 \mathcal{Q}(y_\ell^t) \right)^{-1} = \frac{1}{4 \sum_{j=1}^n p_{\ell,j} w_{\ell,j}^t} \boldsymbol{I}$, by injecting this expression into the above Newton step, we get

$$y_\ell^{t+1} = \frac{\sum_j w_{\ell,j}^t p_{\ell,j} y_j}{\sum_j w_{\ell,j}^t p_{\ell,j}}, \tag{35}$$

which is exactly the repulsion-free fixed-point iteration (25). □

Moreover, the previous theorem also shows that the repulsion-free fixed-point iteration (25) can be obtained as Newton's method.

### 4.3 Connecting the OoS Fixed-point Iteration to Other Methods

Thanks to the form of the repulsion-free fixed-point iteration (25), we can provide connections to other methods in ML, manifold learning, and clustering techniques.

### 4.3.1 Connections to the Mean Shift Algorithm

In the following, we draw connections with the literature on the gradient of the density estimation and its mean shift algorithm for mode seeking. The Parzen (kernel) density estimate from the samples $y_1, y_2, \ldots, y_n$ takes the form

$$\widehat{p}(y) = \sum_{j=1}^n k(\|y - y_j\|^2), \tag{36}$$

for some profile of a kernel $k(\cdot)$, such as the Gaussian and Epanechnikov kernels. A fundamental property of a density function is its modes, namely where it has local maxima. The nullification of gradient of the

density enables one to seek these modes (Fukunaga & Hostetler, 1975). Since the derivative of the density function estimate (36) with respect to $y$ is

$$\nabla_y \widehat{p}(y) = \sum_{j=1}^n \nabla_y k(\|y - y_j\|^2),$$

(37)

its nullification yields the so-called mean shift algorithm with the following fixed-point iteration

$$y^{t+1} = \frac{\sum_j k' \left(\|y^t - y_j\|^2\right) y_j}{\sum_j k' \left(\|y^t - y_j\|^2\right)}.$$

(38)

In this expression, $k'(\cdot)$ is the derivative of the profile of a kernel $k(\cdot)$. The latter is called the shadow of the former. This celebrated mean shift algorithm has been largely investigated for clustering and mode seeking, and theoretical findings have attracted much attention, including guarantees of asymptotic unbiasedness, consistency, and uniform consistency of this estimate (Fukunaga & Hostetler, 1975). More recent studies have addressed theoretical insights such as convergence analysis (Carreira-Perpinan, 2007; Aliyari Ghassabeh, 2013; Cariou et al., 2022; Yamasaki & Tanaka, 2024).

One can view the proposed fixed-point update rule (25) in light of this literature on the gradient of the density and its mean shift algorithm. However, while the mean shift (38) has a single function $k'(\cdot)$, the OoS fixed-point iteration (25) has two weighting functions: The weight $w_{\ell,j}$ defined by the function $w(u) = 1/(1 + u)$ for $u = \|y_\ell^t - y_j\|^2$ plays the same role as $k'(\cdot)$ in (38), and $p_{\ell,j}^t$ measures the affinity between $x_\ell$ and $x_j$. Hence, one can view this as a mean shift that seeks the modes of two joint density estimates, one defined by the shadow of $p_{\ell,j}$ (which is also a Gaussian profile function) in the high-dimensional space $\mathbb{X}$, and one defined by the shadow of $w_{\ell,j}$ (which is the profile function $\log(1 + \|y_\ell^t - y_j\|^2)$) in the lower-dimensional space $\mathbb{Y}$. However, this analogy needs to be carefully handled, since the function $\log(1 + \|y_\ell^t - y_j\|^2)$ is not a valid kernel for density estimation. Moreover, due to its form, the fixed-point update rule (25) is more complex than the mean shift iteration (38). Nevertheless, this perspective provides us with a clustering viewpoint of the proposed fixed-point iteration, which corroborates the theoretical findings on t-SNE as a clustering method (Linderman & Steinerberger, 2019; Cai & Ma, 2022).

### 4.3.2 Connections to the Pre-image Resolution In ML

Another connection can be made with the pre-image problem in ML and its resolution. Many ML frameworks, such as kernel machines and deep learning, aim to map the input data into a more relevant higher-dimensional space $\mathbb{X}$ (often called feature space or latent space). Nevertheless, one often needs to get back to the observation space $\mathbb{Y}$, which requires solving the pre-image problem (Honeine & Richard, 2011), namely estimating the inverse of the nonlinear embedding. The pre-image problem arises in many ML tasks, often when aiming for interpretable models (Tran Thi Phuong et al., 2020). Roughly speaking, for some map $\phi(\cdot)$ from $\mathbb{Y}$ to $\mathbb{X}$, ML frameworks construct a linear model in the latter, namely of the form

$$\psi = \sum_{j=1}^n \alpha_j \, \phi(y_j),$$

(39)

for some coefficients $\alpha_1, \alpha_2, \ldots \alpha_n$ optimized in the latent space for a given criterion (e.g. maximizing the margin in support vector machines, capturing most of the projected variance in kernel PCA). Estimating the pre-image $y$ of $\psi$ can be done by minimizing the norm of the residual between $\psi$ and $\phi(y)$ in $\mathbb{X}$. When dealing with a radial kernel $k(\cdot)$ that induces the embedding $\phi(\cdot)$, we get

$$\Gamma(y) = -\sum_{j=1}^n \alpha_j \, k(\|y - y_j\|^2).$$

(40)

By examining its gradient and nullifying it, we get the following fixed-point iteration

$$y^{t+1} = \frac{\sum_{j=1}^n \alpha_j k'(\|y^t - y_j\|^2) y_j}{\sum_{j=1}^n \alpha_j k'(\|y^t - y_j\|^2)},$$

(41)

where we have explored the structure of the radial kernels, with $k'(\cdot)$ being the derivative of the profile $k(\cdot)$. Moreover, when dealing with the Gaussian kernel, $k'(u) = -ck(u)$ for some normalization constant $c$. For recent theoretical results on the pre-image problem and its resolution with the fixed-point iteration (41), see Honeine (2024).

Even though the perspectives of input versus output spaces are flipped between t-SNE and (kernel-based) ML, there are several connections that can be made between the pre-image problem in ML and the OoS extension of t-SNE. Both share embedding data from a high-dimensional space to a low-dimensional one, and both operate OoS extensions. Moreover, we can provide some analogies on their fixed-point iteration schemes (25) and (41), in the same spirit as previously conducted with the mean shift analogy. However, as in the previous discussion, such analogy needs to be handled with care for the same reasons, as $k'(\cdot)$ can correspond to $w_{\ell,j}$ if $k(u) = \log(1 + u)$, which is not a valid positive definite kernel. Still, the pre-image viewpoint is interesting as one can directly correspond the coefficients $\alpha_1, \alpha_2, \ldots \alpha_n$ to the affinity entries $p_{\ell,1}, p_{\ell,2}, \ldots, p_{\ell,n}$, both being computed in the high-dimensional space.

## 5 Experiments

This section provides extensive experimental results that evaluate the proposed methods. For this purpose, the experiments were conducted on three real data sets: MNIST (LeCun & Cortes, 2010), Fashion MNIST (Xiao et al., 2017) and the Mouse Retina data sets (Macosko et al., 2015). After introducing the experimental settings in Section 5.1, Section 5.2 describes in detail the conducted methodology with results on MNIST, while Section 5.3 extends the experimental analysis to the data sets Fashion MNIST and Mouse Retina. Moreover, we provide in-depth experimental analysis in Section 5.4 to study robustness to low-density data, algorithmic sensitivity, and robustness to out-of-distribution samples.

### 5.1 Experimental Settings

We generated the t-SNE reference embedding using the same settings as proposed by van der Maaten & Hinton (2008) with the Python implementation in scikit-learn and its default parameters (Pedregosa et al., 2011), namely with perplexity set to 30, early exaggeration set to $\alpha = 4$ and learning rate $\eta = 1'000$. In all experiments, we selected randomly $m = 30$ OoS samples, namely not from the $n$ samples used in the t-SNE training. For the analysis of the OoS methods, the initial values were sampled from a normal distribution with zero mean and variance 100 in all experiments (initialization sensitivity is studied in Section 5.4.2). For gradient descent algorithms, we used a learning rate value of $\eta^t = 0.1n$, as corroborated by several studies to be of the order of the number of samples (Linderman & Steinerberger, 2019). Other scaling factors as multiples of $n$ were also examined, mainly fractions because $\eta^t = n$ and other multiples do not converge in most cases. The computations were executed on a MacBook Pro with an Apple M1 Max chip and 64 GB memory, with scikit-learn version 1.6.1.

In order to provide a comparative analysis, we explore several resolution methods, grouped in two categories:

- The first one covers methods that consider both attractive and repulsion terms in the optimization. This includes the gradient descent on the non-symmetrized variant of the KL divergence, as presented in (10) and studied by Poličar et al. (2023), the gradient descent on the symmetrized variant of the augmented gradient (16), as well as the associated fixed-point iteration (17).

- The second category involves methods that do not take into account the repulsion term in the optimization. Such repulsion methods include the repulsion-free fixed-point iteration (25), as well as gradient descent variant without the repulsion term in (16).

### 5.2 Experimental Results on the MNIST Data Set

MNIST is a well-known real data set of $28 \times 28$-pixel grayscale images of handwritten digits of the 10-digit classes ("0" through "9"). We used the same configuration as proposed by van der Maaten & Hinton (2008) in order to be comparable with the main results in the literature. The total number of images, initially $70'000$

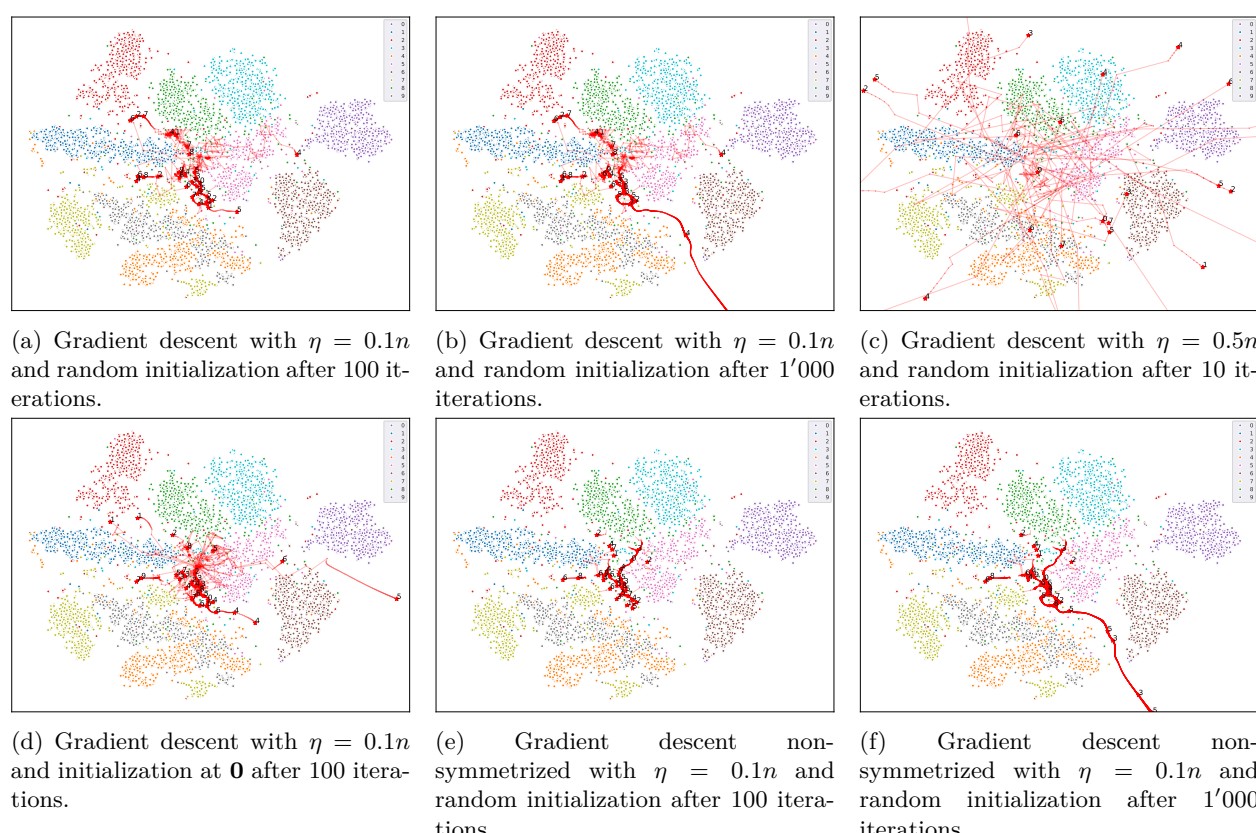

(a) Gradient descent with $\eta = 0.1n$ and random initialization after 100 iterations.

(b) Gradient descent with $\eta = 0.1n$ and random initialization after $1'000$ iterations.

(c) Gradient descent with $\eta = 0.5n$ and random initialization after 10 iterations.

(d) Gradient descent with $\eta = 0.1n$ and initialization at **0** after 100 iterations.

(e) Gradient descent non-symmetrized with $\eta = 0.1n$ and random initialization after 100 iterations.

(f) Gradient descent non-symmetrized with $\eta = 0.1n$ and random initialization after $1'000$ iterations.

Figure 2: Results of the gradient descent algorithms on MNIST. The t-SNE embedding of the $n = 6'000$ images in colored dots for the 10 classes of digits. The estimated embeddings of the 30 OoS images are given in red stars ($\star$) with their respective labels next to them, and the solution paths are indicated by red lines ($-$).

images, was therefore reduced to $n = 6'000$ images. The images in MNIST were treated as 784-dimensional vectors. In the following, we assess 2D embedding by examining qualitative and quantitative criteria.

### 5.2.1 Qualitative Results

We first study the gradient descent algorithms with both attraction and repulsion terms for several parameters. Figure 2 illustrates the t-SNE embedding of the $n = 6'000$ samples and the obtained estimates for 30 OoS images. The main result is in the upper left figure, obtained after 100 iterations with a learning rate of $\eta = 0.1n$ and random initialization (Figure 2a). The other figures illustrate the effects of an increased number of iterations to $1'000$ (Figure 2b), a learning rate of $0.5n$ (Figure 2c), and initialization at the origin **0**. Similar results are obtained using the non-symmetrized variant of the gradient descent algorithms. All these results show the gradient descent scheme fails to position well the OoS samples on the background t-SNE map. We presume that this is mainly due to the attraction-repulsion terms in the gradient, as OoS samples tend to occupy the gap between the clusters of the reference t-SNE embedding.

Since the attraction-repulsion terms in the gradient (16) seem to operate mainly by repulsion from the reference t-SNE data, we study the effect of reducing the impact of the repulsion. This can be done by increasing the exaggeration, set to the default value $\alpha = 1$ in Figure 2. Figure 3 illustrates the results obtained for values of $\alpha = 100$ and $\alpha = 1'000$. Although several OoS images are mapped to the relevant clusters for $\alpha = 100$, increasing to $1'000$ iterations does not provide significant improvement. One could get better results by increasing the exaggeration to $\alpha = 1'000$, but this requires a fine-tuning of the learning rate (e.g., $\eta = 0.01n$), as given for the gradient descent in Figure 3c and its non-symmetrized variant in

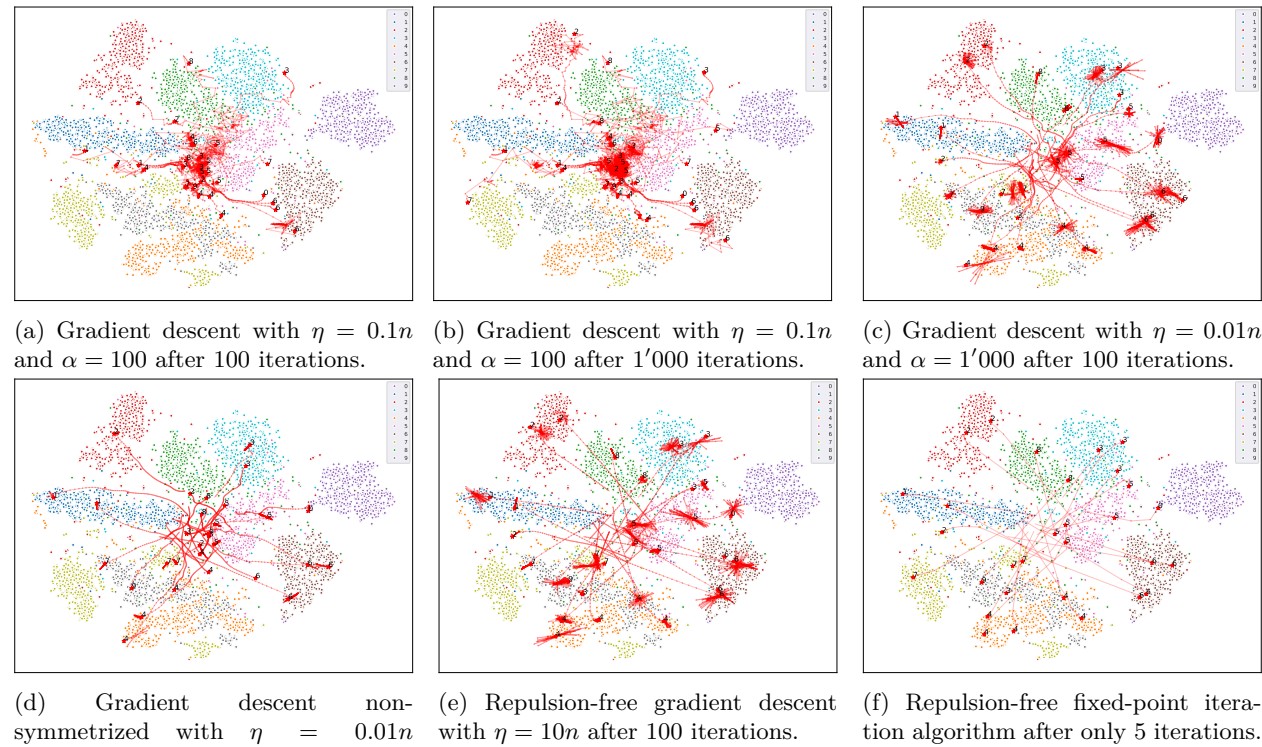

(a) Gradient descent with $\eta = 0.1n$ and $\alpha = 100$ after 100 iterations.

(b) Gradient descent with $\eta = 0.1n$ and $\alpha = 100$ after $1'000$ iterations.

(c) Gradient descent with $\eta = 0.01n$ and $\alpha = 1'000$ after 100 iterations.

(d) Gradient descent non-symmetrized with $\eta = 0.01n$ and $\alpha = 1'000$ after 100 iterations.

(e) Repulsion-free gradient descent with $\eta = 10n$ after 100 iterations.

(f) Repulsion-free fixed-point iteration algorithm after only 5 iterations.

Figure 3: Results of different exaggeration values ($\alpha$) for the gradient descent and fixed-point iteration algorithm on MNIST. Same legend as Figures 2.

Figure 3d. We can also observe that the non-symmetrized variant fails to properly position many OoS images, compared to the vanilla variant for the same parameters. An easier strategy is through the repulsion-free gradient descent algorithm, where only the learning rate needs to be tuned, as given in Figure 3e for $\eta = 10n$, which can also be viewed as $\eta = 0.01n$ with an exaggeration of $\alpha = 1'000$.

We also examine the fixed-point iteration algorithms, where the only tunable parameter is the exaggeration value, set by default to $\alpha = 1$. We consider the three variants: The fixed–point with both attraction and repulsion terms (17), its non-symmetrized variant (18), and the repulsion-free variant (25). The experimental results show that both variants of the attraction-repulsion fixed-point iteration algorithm, with or without symmetrization, suffer the same weaknesses as their gradient-based counterparts, as described in the previous paragraphs. By increasing the exaggeration value in order to enhance the results, one ends up eventually with the repulsion-free fixed-point iteration algorithm (25). Figure 3f provides the results of the repulsion-free fixed-point iteration after only 5 iterations, illustrating its efficiency to embed the OoS images in their respective clusters. This qualitative assessment is supported by the quantitative evaluation using the KL divergence, trustworthiness and continuity, as provided in the following.

### 5.2.2 Evaluation of the KL Divergence

To provide a quantitative evaluation of the methods, we compute the KL divergence over all the data, namely the KL divergence (12) augmented with all $m = 30$ OoS estimates. We expect small variations between the different methods, since more than 99.5% of the used data are the fixed training data. Nevertheless, the augmented KL divergence provides a quantitative assessment of the resolution of the OoS problem, highlighting the relevance of the working direction and its convergence.

Figure 4 shows the evolution of the KL divergence for several methods and parameters. The compared methods consider non-symmetrized vs symmetrized affinities in the optimization problem, variants with or

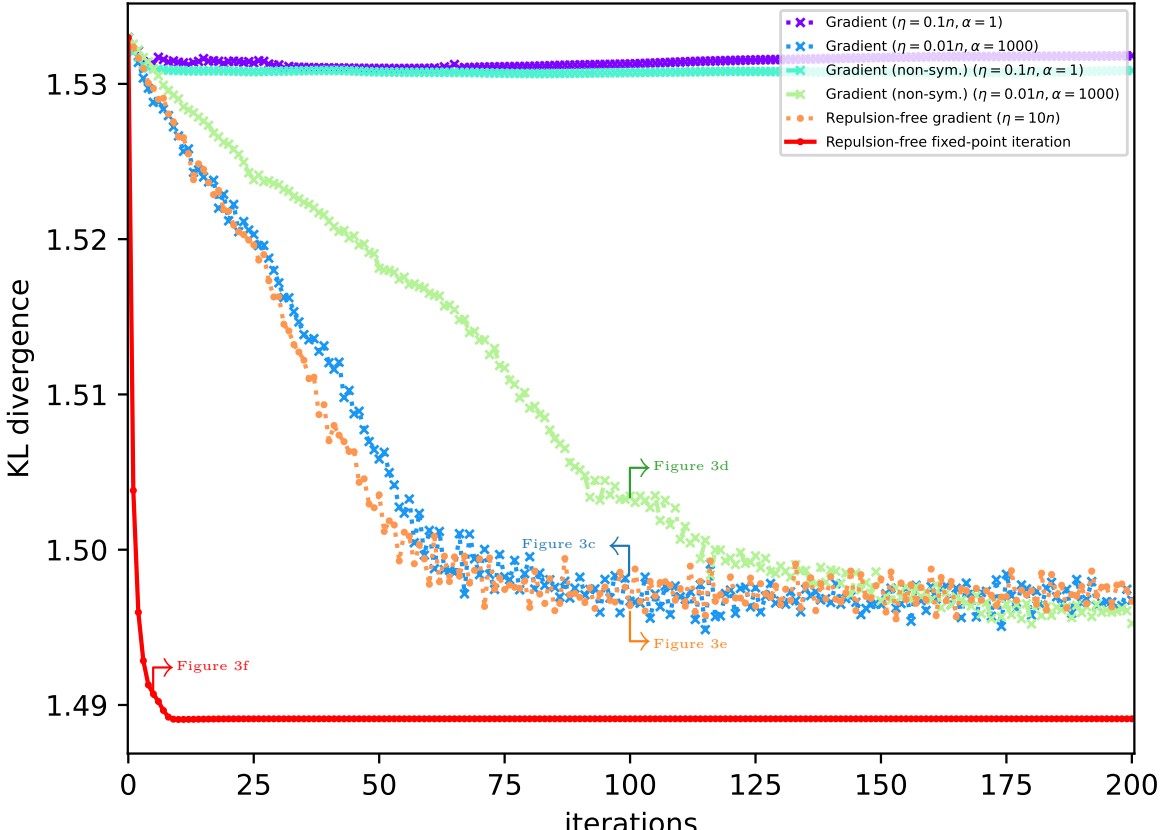

Figure 4: Evolution of the augmented KL divergence over the first 200 iterations for different methods and parameters. These methods are grouped in two categories: Methods relying on both attraction and repulsion (marker ✕), and those without the repulsion (marker ●). We also distinguish between methods based on the non-symmetrized probability (dashdoted lines ▬ ·) from the ones using the symmetrized affinities, either using a gradient descent (dotted lines ·· ·) or a fixed-point iteration (solid lines ▬).

without the repulsion term, and optimization using gradient descent vs fixed-point iteration algorithms. Moreover, the parameters were selected from the best qualitative visual results obtained in the previous figures. These results demonstrate the high performance of the proposed repulsion-free methods, gradient-based and fixed-point iteration algorithms. Moreover, the latter converges faster within only a couple of iterations, and without the cumbersome fine-tuning of any parameter.

### 5.2.3 Trustworthiness and Continuity

The evaluations conducted so far either consider the qualitative evaluation through the 2D embedding or quantitatively in terms of the augmented KL divergence. In contrast to these analyses, we consider in the following a data-based metric, by evaluating to what extent the local neighborhoods are preserved in both spaces. This is measured with two metrics, Trustworthiness and Continuity (Venna & Kaski, 2006; Van Der Maaten, 2009).

Trustworthiness is a metric that measures to what extent, if the low-dimensional embeddings of two samples are close neighbors, they should be close to each other in the original space. We consider the following

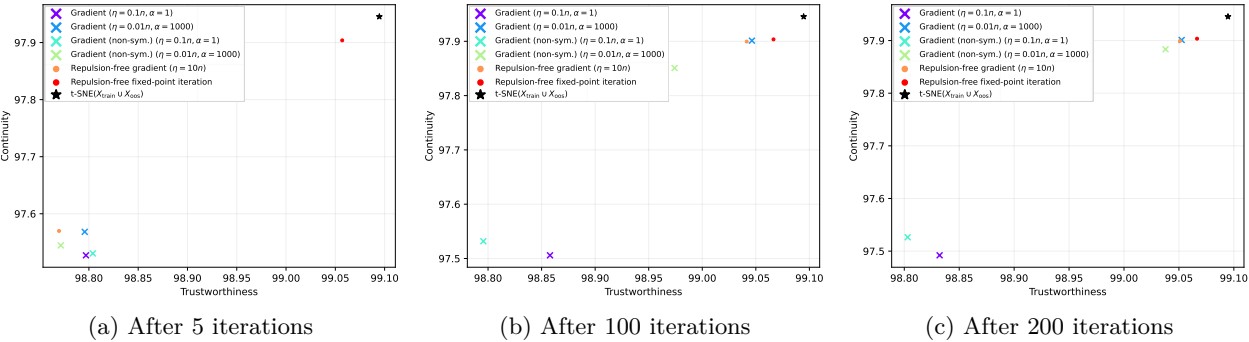

|||
|(a) After 5 iterations|(b) After 100 iterations|(c) After 200 iterations|

Figure 5: Trustworthiness and Continuity for different methods on MNIST. These metrics were evaluated after 5 iterations (left), after 100 iterations (center), and after 200 iterations (right). Same markers and colors as Figure 4. Moreover, ★ denotes the best-case reference obtained from a t-SNE trained jointly on all reference and OoS images.

formulation for $k$ nearest neighbors applied on a set of $n$ samples (Pedregosa et al., 2011):

$$T(k) = 1 - \frac{2}{nk(2n - 3k - 1)} \sum_{i=1}^{n} \sum_{y_j \in \mathcal{N}_k(y_i)} \max\left\{0, \rho_{x_i}(x_j) - k\right\}, \tag{42}$$

where $\mathcal{N}_k(y_i)$ denotes the $k$ nearest neighbors of $y_i$ in the output space, and $\rho_{x_i}(x_j)$ is the rank order of the data sample $x_j$ from the $x_i$, namely in the original space. In other words, the penalty assigned to unexpected local neighbors in the embedded space is proportional to their rank in the original space. With values between 0 and 100%, values around 0.5 could correspond to the worst case with random neighborhoods, but could be higher for the worst possible practical results (Kaski et al., 2003). The complement of Trustworthiness is Continuity, where neighboring samples in the original space should be projected close to each other in the low-dimensional space; otherwise, there are discontinuities in the projection. Its expression is similar to (42), where the two spaces are interchanged.

To provide a comprehensive analysis, we examine both Trustworthiness and Continuity metrics by evaluating them on the set of all the data, namely $n$ training and $m$ OoS samples, allowing us to assess the overall embedding. Since this evaluation includes the t-SNE reference data, which constitutes up to 99.5% of the data, we can expect very high values with low variations between the methods. Figure 5 provides the Trustworthiness and Continuity metrics for the MNIST data set, evaluated after 5 iterations (Figure 5a), after 100 iterations (Figure 5b), and after 200 iterations (Figure 5c). In these figures, we have also included the best-case reference, given by ★, which results from a t-SNE trained jointly on all the data, namely, $n$ reference and $m$ OoS samples. The obtained results corroborate the KL divergence results given in Figure 4. They demonstrate that the repulsion-free fixed-point iteration algorithm preserves the vicinity in both spaces in only 5 iterations, with results very close to the best-case reference. The other methods require a meticulous fine-tuning of their parameters to provide comparative results, but with a larger number of iterations. Moreover, the non-symmetrized variant lags behind with even more iterations.

### 5.3 Experimental Results on Fashion MNIST and Mouse Retina Data Sets

In this section, we extend the experimental analysis described in the previous section to two other real data sets: Fashion MNIST and Mouse Retina data sets.

#### 5.3.1 Fashion MNIST Data Set

Fashion MNIST is a well-known data set of $28 \times 28$-pixel grayscale images of fashion items grouped into 10 classes: T-shirt/Top, Trouser, Pullover, Dress, Coat, Sandal, Shirt, Sneaker, Bag, Ankle boot (Xiao et al., 2017). It is well-known from the literature that this is a more complex data set by nature, with several clusters of mixed classes. For this reason, we considered $n = 10'000$ images.

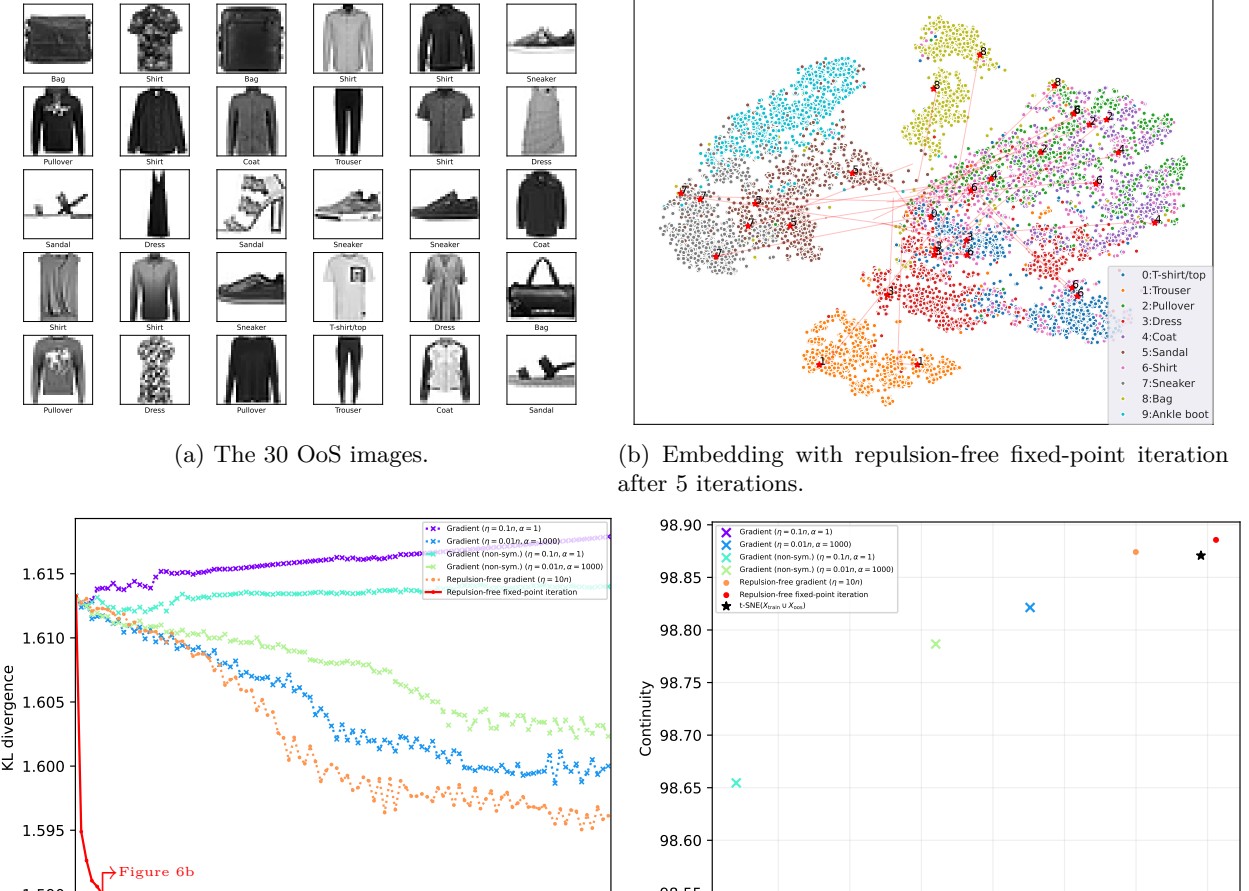

(a) The 30 OoS images.

(b) Embedding with repulsion-free fixed-point iteration after 5 iterations.

(c) KL divergence over 100 iterations.

(d) Trustworthiness and Continuity after 100 iterations.

Figure 6: Results on the Fashion MNIST data set. The first row shows the 30 OoS images (upper left) and their embedding on the reference t-SNE (upper right) using the repulsion-free fixed-point iteration algorithm after 5 iterations. The lower row provides a comparative analysis of the evolution of the KL divergence (lower left) and the Trustworthiness and Continuity metrics (lower right) for several methods. Same legends as Figures 2 and 4.

We considered the same settings and experimental analysis as for the MNIST. Figure 6a shows the 30 OoS images used to evaluate the proposed methods. Figure 6b shows the t-SNE reference and the estimated embedding of the 30 OoS images after only 5 iterations using the proposed repulsion-free fixed-point iteration method. A comparative analysis is provided in the second row of Figure 6, with the learning curves in terms of the KL divergence estimated on all the data (Figure 6c) and the Trustworthiness and Continuity after 100 iterations (Figure 6d). The obtained experimental results corroborate the results obtained on MNIST, demonstrating the relevance of our work on this complex data set, as some clusters are not well separated by t-SNE. Moreover, the repulsion-free fixed-point iteration algorithm outperforms the other methods, and even outperforms the best-case reference obtained from a t-SNE trained jointly on all reference and OoS images.

### 5.3.2 Mouse Retina Data Set

It is well known that in t-SNE, the cluster area is proportional to its ratio of the number of training data. Therefore, t-SNE and its OoS extensions could suffer when addressing data sets that have a large imbalance

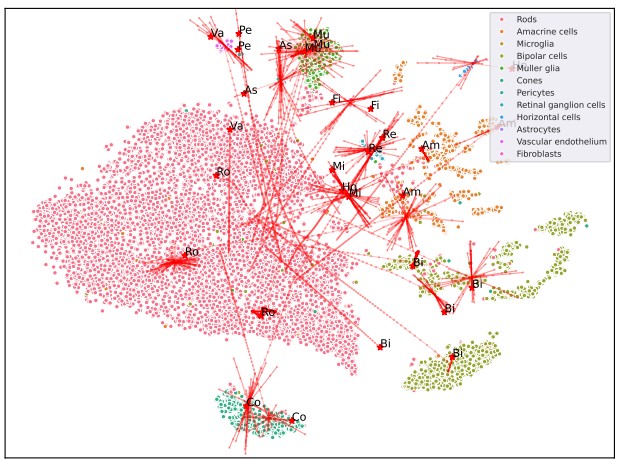

(a) Embedding with repulsion-free gradient descent with $\eta = 10n$ after 100 iterations

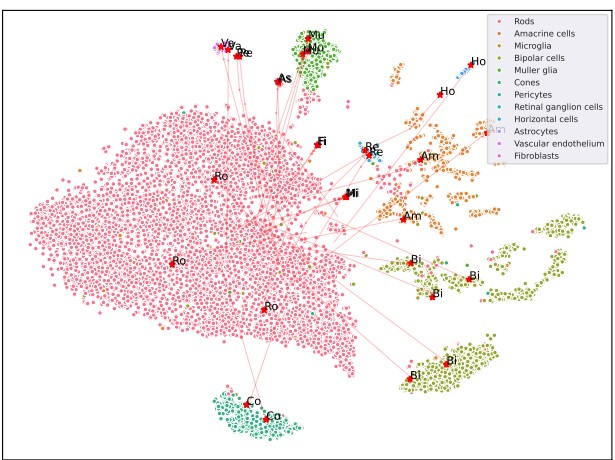

(b) Embedding with repulsion-free fixed-point iteration after 5 iterations

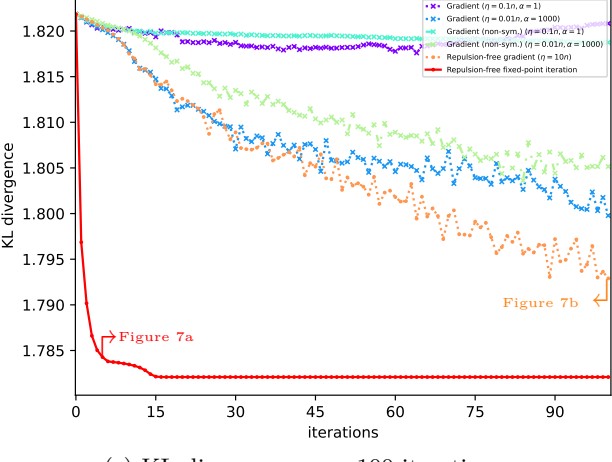

(c) KL divergence over 100 iterations.

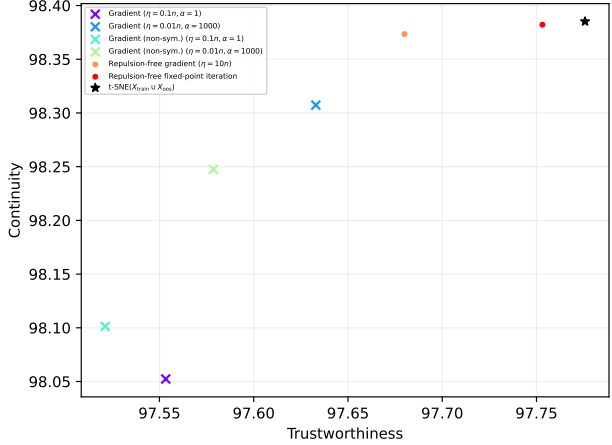

(d) Trustworthiness and Continuity after 100 iterations

Figure 7: Results on the Mouse Retina data set, which suffers from imbalanced classes. The first row shows embeddings using the repulsion-free gradient descent algorithm after 100 iterations (upper left) and its fixed-point counterpart after only 5 iterations (upper right). The lower row provides a comparative analysis for several methods of the evolution of the KL divergence (lower left) and the Trustworthiness and Continuity metrics after 100 iterations (lower right). Same legend as Figures 2 and 4.

in their classes. In this section, we aim to study the sensitivity of the proposed methods to such imbalance. For this purpose, we consider the Mouse Retina data set from (Macosko et al., 2015) (see also Poličar et al. (2023)), which constitutes an atlas of mouse retinal cell types based on the Drop-seq protocol. The total data set has $44,808$ samples with 50 features belonging to 12 cell types. This data set was chosen because it has a major imbalance between the classes, with more than half of the samples belonging to a single class, "Rods" with $29'000$ samples, the second largest being "Bipolar cells" with $6,285$ samples, ... and the three smallest classes each have less than 70 samples. We consider a subset of $10'000$ training data with the same class proportion as in the original data set, as well as 30 OoS samples. To analyze the effect of such high imbalance, each of the 12 classes are represented by at least 2 OoS samples.

The obtained results given in Figure 7 demonstrate the high performance of the proposed repulsion-free fixed-point method, even in such a problem of imbalanced classes, followed by the repulsion-free gradient. This illustrates the relevance of these methods to deal with data suffering from highly imbalanced classes.

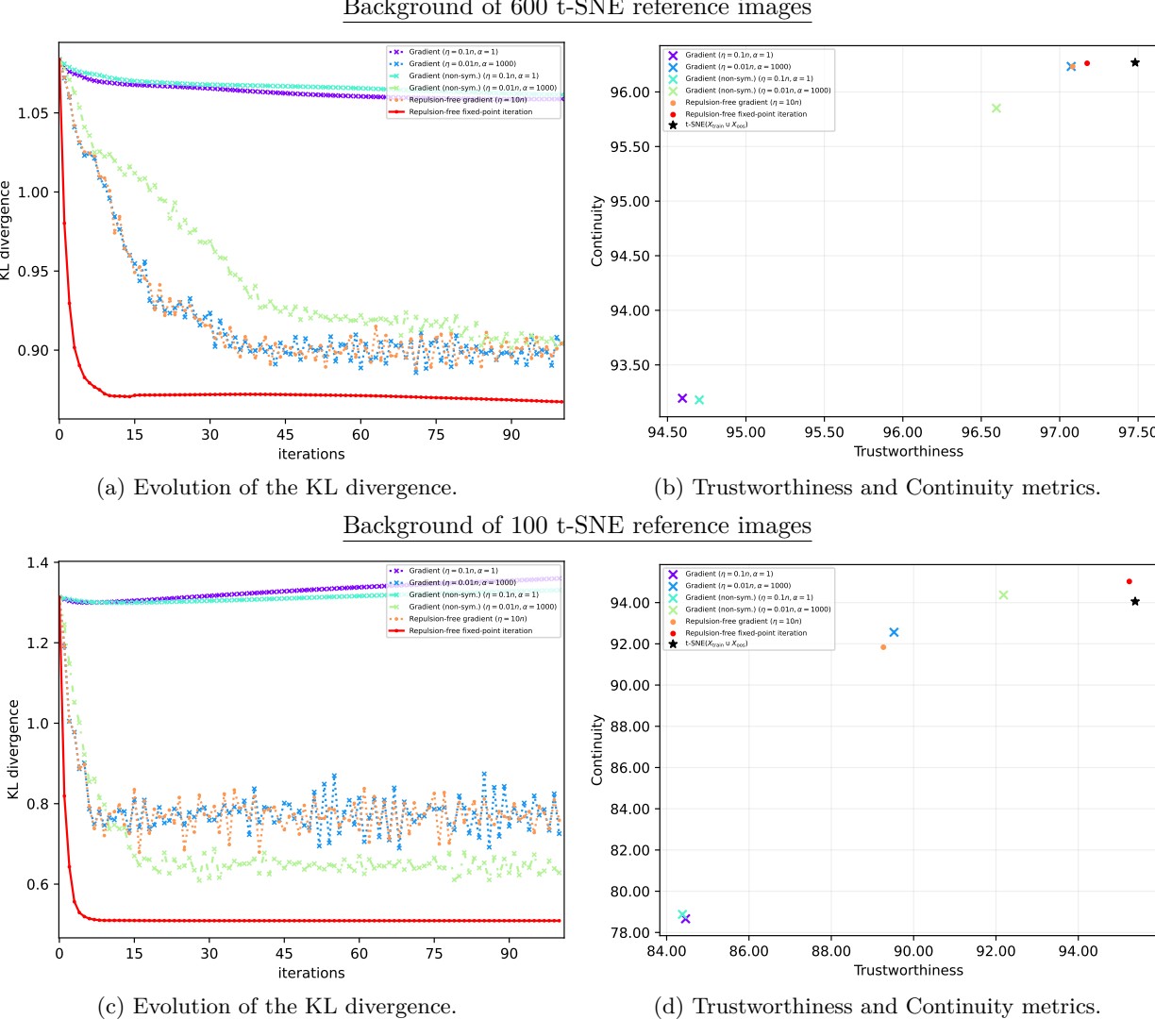

Figure 8: Sparse data analysis. Results obtained on low-density data, with only 600 (upper figures) and 100 (lower figures) t-SNE reference images of MNIST. Results are given in terms of KL divergence, Trustworthiness and Continuity for different methods after 100 iterations.

## 5.4 Robustness Analysis

Following the sensitivity analysis to class imbalance conducted in the previous section, we examine in this section other aspects of robustness: the robustness in addressing low-density samples, the algorithmic sensitivity (e.g., effect of different initializations) and the generalization ability for out-of-distribution samples.

### 5.4.1 Sparse Data Analysis

We study the issue of having sparse data living in a high dimensional space, thus suffering from the curse of dimensionality. For this purpose, we consider a subset of MNIST with only $n = 600$ t-SNE training samples, and go further with only $n = 100$ samples. These cases correspond to critical situations of having less images than the dimension of the data (i.e., images of 784 pixels). The results illustrated in Figure 8 show that several of the proposed methods still give relevant results in such challenging conditions, where the metrics used are the KL divergence and both Trustworthiness and Continuity. The proposed repulsion-free

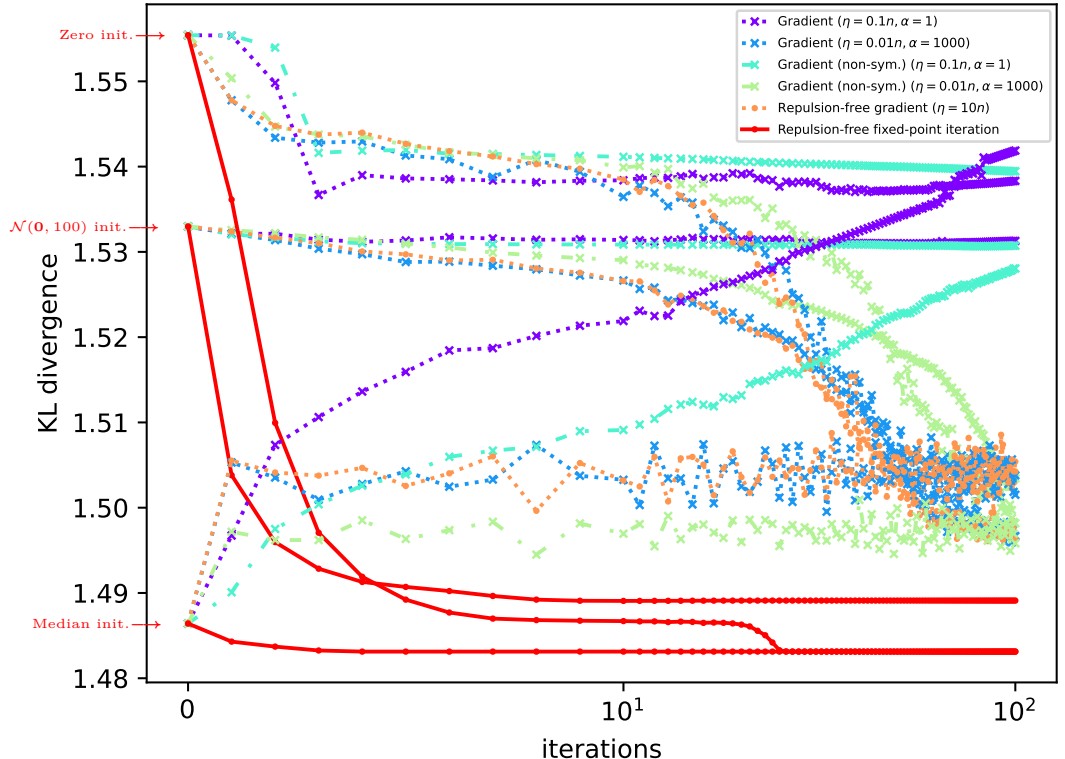

Figure 9: Sensitivity to initialization. Evolution of the KL divergence over 100 iterations for different OoS methods and settings, starting from three different initializations: zero, random distribution and median-based heuristic.

fixed-point iteration algorithm still outperforms the other methods, followed by the gradient descent with appropriate fine-tuning.

### 5.4.2 Algorithmic Sensitivity Analysis

One can easily integrate a momentum acceleration into all the methods presented in this paper, not only the gradient descent but also the fixed-point iteration algorithms. However, preliminary experiments showed that the momentum has a low impact on the solution while requiring cumbersome fine-tuning. For these reasons, we did not provide an analysis of the usage of a momentum acceleration.

**Sensitivity to Initialization**

In the following, we provide an initialization sensitivity study, since the initialization is the only choice to consider for the fixed-point iteration algorithms. For this purpose, we study three different initializations:

- Random initialization. It consists of initializing by sampling from a normal distribution with zero mean and standard deviation of 100. This initialization was considered in all the previous experiments (except otherwise specified as given in Figure 2d for comparative analysis).

- Zero initialization. Here, $y_\ell^0 = \mathbf{0}$ for all $\ell = 1, 2, \ldots, m$. One could also initialize with any other fixed location, such as one of the four corners of the t-SNE reference map, since the corners are seldom occupied in general.

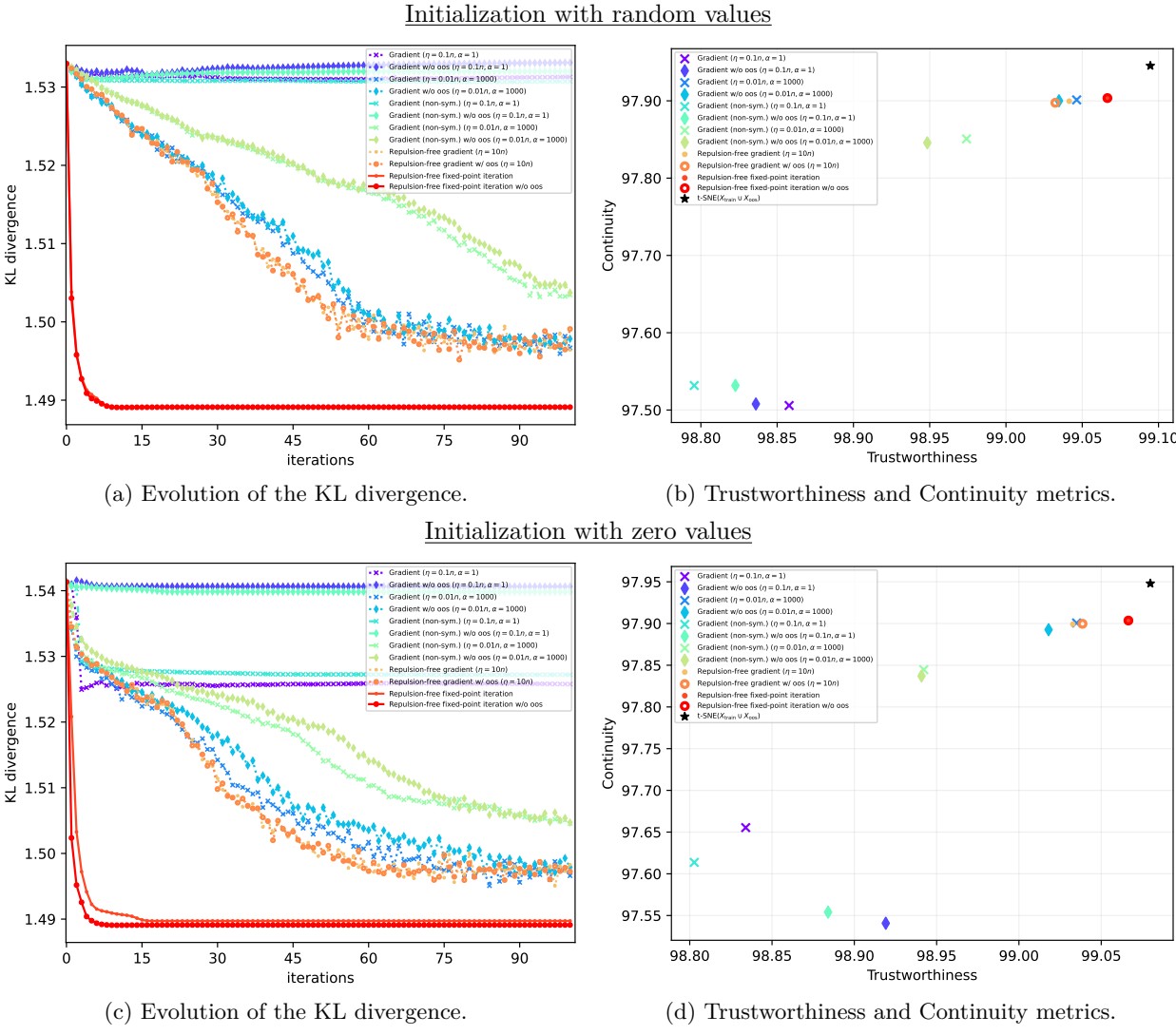

Figure 10: Effect of integrating vs excluding the other OoS samples in the optimization. The obtained results are given for the random initialization (upper figures) and the zero initialization (lower figures), where each figure presents the gradient descent and fixed-point iteration algorithms with all OoS samples (markers ⚹ and ●, respectively), and without the other OoS samples (markers ◆ and ⬤, respectively).

- Median-based heuristic. It consists of first identifying, for each OoS, its $k$-nearest neighbors in the high-dimensional space, and then positioning its embedding in the low-dimensional space as the median of the t-SNE embeddings of these $k$ references. The median-based heuristic was used for the t-SNE application on the single-cell transcriptomics by Kobak & Berens (2019).

Figure 9 shows the effect of the three initializations on the KL divergence and its evolution over the first 100 iterations for the different OoS methods and settings. In the following, we focus on the repulsion-free fixed-point iteration algorithm, since it is the only one that reduces the KL divergence in all initialization cases. We can see that the zero initialization, which starts at a higher value of the KL divergence, still converges fast to a better state. Finally, the initialization with the median-based heuristic, which was used as a warm start in (Poličar et al., 2023), allows indeed to start at a low KL divergence value. However, the gradient descent methods fail to further reduce it. The only exception is the repulsion-free fixed-point iteration algorithm, which further minimizes the KL divergence.

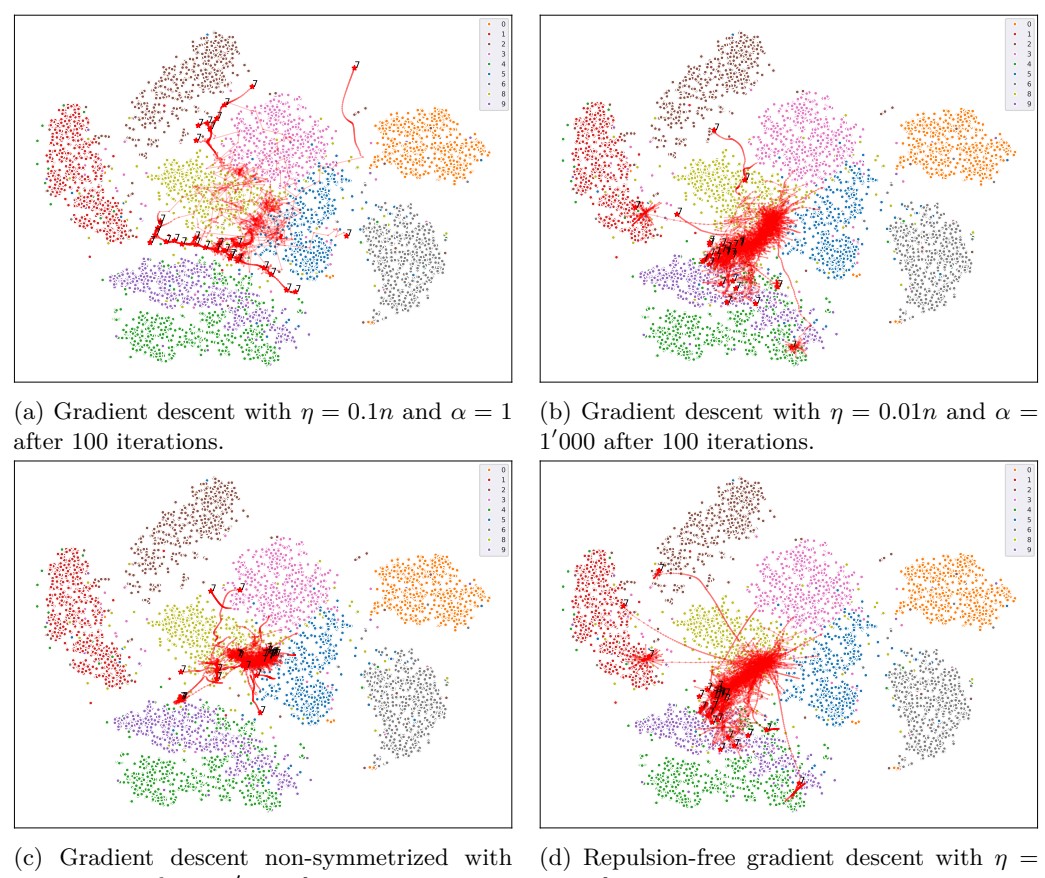

(a) Gradient descent with $\eta = 0.1n$ and $\alpha = 1$ after 100 iterations.

(b) Gradient descent with $\eta = 0.01n$ and $\alpha = 1'000$ after 100 iterations.

(c) Gradient descent non-symmetrized with $\eta = 0.01n$ and $\alpha = 1'000$ after 100 iterations.

(d) Repulsion-free gradient descent with $\eta = 10n$ after 100 iterations.

Figure 11: Analysis of the out-of-distribution robustness for gradient descent methods. These results were obtained when considering the OoS from the class "7", which was excluded from the t-SNE reference. Same legend as Figures 2.

**Sensitivity to Multiple OoS Optimization**

Finally, we analyze the effect of integrating all the OoS within the optimization problem, and therefore its resolution. We compare the two strategies introduced in Section 3.3: Independent optimization strategy (Section 3.3.1) versus joint optimization strategy (Section 3.3.2). In the former, the $\ell$-th OoS embedding is computed while excluding the other $m - 1$ OoS samples from the minimization of the KL divergence (12), and therefore from all the following expressions, the estimation of $y_\ell^{t+1}$ no longer depends on the estimates of the other OoS samples, but only on the t-SNE reference. This would enhance the convergence, as the results depend only on the fixed background. However, the price to pay for this improvement is that, by solving it independently from the other OoS samples, we lose the effect of attraction between them, and thus clustering. This is demonstrated for the repulsion-free fixed-point iteration algorithm in Figure 12. Figure 10 provides a comparative analysis of the effect of including vs excluding the other OoS samples from the optimization when considering the MNIST task studied in Section 5.2, namely using 30 OoS samples from the 10 classes. These results show that integrating vs excluding the OoS samples has a small effect on the results in terms of KL divergence, Trustworthiness and Continuity measures.

### 5.4.3 Robustness to Out-of-distribution Samples

To study the robustness to out-of-distribution OoS samples, the explored protocol corresponds to a leave-one-class-out, where t-SNE reference background relies on data from all the classes except one, the one that constitutes the OoS class. For this purpose, we examine the MNIST data set, by training the t-SNE

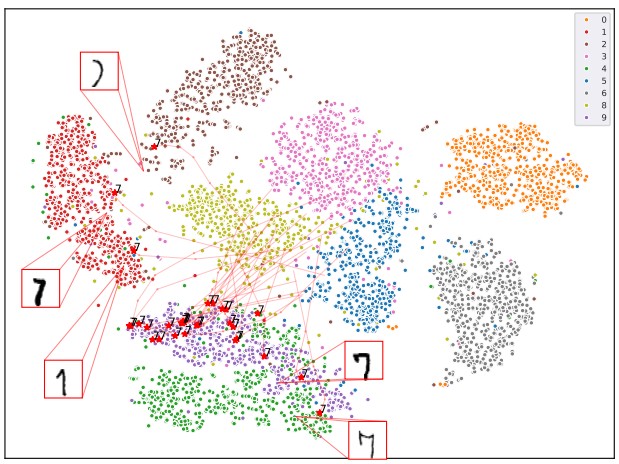 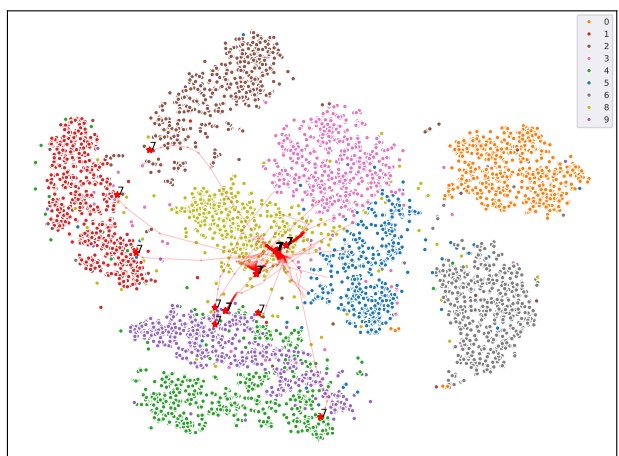

(a) Repulsion-free fixed-point iteration algorithm by excluding the other OoS samples. Results after 5 iterations.

(b) Repulsion-free fixed-point iteration algorithm by including the other OoS samples. Results after 100 iterations.

Figure 12: Analysis of the out-of-distribution robustness of the repulsion-free fixed-point iteration methods. Same settings as Figure 11. The left figure illustrates the results obtained after 5 iterations of the algorithm that excludes the other OoS samples from the optimization. We also highlight several OoS images that are mapped closer to the cluster of digits "1" (the two left images) or to the cluster of digits "2" (the upper image) or between the clusters of digits "4" and "9" (the two right images). The right figure illustrates the results obtained by including the other OoS samples in the optimization.

embedding on $n$ samples from 9 classes, excluding the class of the digit "7" that constitutes the class of the "30" OoS images.

Figure 11 illustrates the results obtained from gradient descent algorithms after 100 iterations. The obtained results are roughly similar to the ones obtained in Figures 2 and 3 in terms of occupying the gap between the clusters. A joint fine-tuning of the learning rate and exaggeration allows us to have a major cluster of the OoS samples. The resulting major cluster is at the vicinity of cluster "9" and close to cluster "4", as expected from the t-SNE maps obtained from all the classes (see Figures 2 and 3). However, the non-symmetrized variant provides less relevant results.

We examine in the following the results in Figure 12 obtained using the repulsion-free fixed-point iteration algorithm. Figure 12b shows that the repulsion-free fixed-point iteration algorithm does not give relevant results. Although some of the OoS samples are well mapped to clusters of related forms (see their forms in the right-hand-side figure), it fails to clearly separate the other samples as many estimates become aggregated. This is due to the missing repulsion term in the optimization, as these estimates attract each other and tend to aggregate. To overcome this issue, we propose to exclude the other OoS samples from the optimization, as introduced in Section 3.3.1. In other words, when minimizing the KL divergence (12) for the $\ell$-th OoS sample, we exclude all the other OoS samples from this expression, and therefore from all the following expressions. By doing so, the estimation of $y_\ell^{t+1}$ no longer depends on the estimates of the other OoS samples, but only on the t-SNE reference background. The relevance of this approach is illustrated in Figure 12a obtained after only 5 iterations. Most of the samples were mapped to the vicinity of digits "4" and "9", with the only exception being 2 images within the cluster of digits "1" and 1 image within the cluster of digits "2". These results are consistent with the forms of these 3 images.

## 6 Conclusion and Final Comments

In this paper, we studied the problem of the OoS extension of t-SNE. For this purpose, we showed that it is easy to derive algorithms to extend t-SNE to the OoS paradigm. Several optimization methods were

proposed, using either a gradient descent scheme or a fixed-point iteration scheme. The relevance of these methodological contributions was supported by providing major theoretical foundations that allowed us to understand well the underlying optimization mechanism of the fixed-point iteration scheme. Finally, experimental results on well-known data sets demonstrated the relevance of our work, and the superiority of the repulsion-free fixed-point iteration compared to the gradient descent algorithms.

It is surprising that researchers have seldom tackled the OoS extension of t-SNE, although t-SNE remains by far the most cited dimensionality reduction method (de Bodt et al., 2025, Figure 2). Although the available methods from the literature explore hybrid and surrogate resolutions, such as an SNE-like non-symmetrization (Poličar et al., 2023) or the Nelder-Mead Simplex algorithm, we demonstrated the ease of providing the gradient descent scheme. Moreover, we provided further developments by considering a simple fixed-point iteration update rule with many appealing properties. While there are some studies that consider solving the t-SNE embedding with a fixed-point iteration, they do not rely on any theoretical foundations and do not address the OoS extension. Moreover, we proposed in this paper connections to both density estimation with its mean shift algorithm for mode seeking, and the pre-image problem in ML with its fixed-point iteration resolution. These results are also aligned with theoretical findings on t-SNE (Shaham & Steinerberger, 2017; Linderman & Steinerberger, 2019). We think that major results can be obtained by bridging the gap between these frameworks and t-SNE. Moreover, we revealed many interesting properties of the proposed fixed-point iteration.

The proposed repulsion-free variant suffers from some weaknesses. It results from modifying the gradient by removing the repulsion term. A more principled approach would be to change the cost function. Moreover, we did not provide any proof demonstrating that the fixed-point iteration is a contraction mapping, which is a foundational property for convergence guarantees. However, comprehensive experiments showed that the proposed method outperformed the other methods with qualitative and quantitative assessments, including the efficient reduction of the KL divergence and the neighborhood preservation using two data-centric metrics (trustworthiness and continuity). Finally, the motivation for discarding the repulsion term may also find its roots in other well-known methods that do not use repulsion but only attraction, such as Laplacian Eigenmaps, LLE, and many other neighborhood-preserving methods that consider only the local attractions. Furthermore, researchers such as Böhm et al. (2022) have been studied the t-SNE and the UMAP without a repulsion term (i.e., UMAP without the so-called negative sampling).

Still, we think that t-SNE and its OoS extension have not taken full advantage of the theory underlying fixed-point strategies (Combettes & Pesquet, 2021). While this paper focuses on the OoS extension of t-SNE, we think that t-SNE can also benefit from the proposed methodology (namely, the fixed-point iteration) and its solid theoretical foundations. Finally, another future work is solving t-SNE in an online learning strategy, which would integrate not only OoS but also training t-SNE sequentially on the OoS.

## Acknowledgments and Disclosure of Funding

This work was supported by the French *Agence Nationale de la Recherche* (ANR) through the grants ANR-18-CE23-0014 and ANR-23-CE23-0004.

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
