# OpenReview forum: "The Out-of-sample Extensions of t-SNE: From Gradient Descent to Fixed-point Iteration Algorithms"
_TMLR — Accepted by TMLR_

### Review · Reviewer_5crs · 2026-03-04

**Summary Of Contributions:**

The paper presents an extension of the t-SNE dimension reduction technique to embed new samples into the reduced space after the initial training, using a similar formulation to the original t-SNE training method. The proposed method is interesting for its simplicity but its empirical evaluation is too limited at the moment to provide the evidence that it actually works.

**Audience:**

No

**Audience Explanation:**

As stated above, the findings are at the moment not backed by enough empirical evidence

**Broader Impact Concerns:**

None to remark. Being able to visualize data in low-dimensional space is a very relevant task

**Claims And Evidence:**

No

**Claims Explanation:**

At multiple points in the text, some terms that have a precise mathematical meaning seem to be used informally.
In section 2.2, is d(.,.) a distance or a divergence? This is unclear. Are the two distributions histograms over the set of data, or distributions in the input space? Or in the low-dimensional space?
In the experiments, the text refers to trustworthiness and continuity as "dual" of each other. Duality has specific meanings in convex analysis and optimization and should not be overloaded as a synonym for "reverse" or "complement", especially since the function T(.) is only one way among other of expressing the concept as described in the text (proximity in output means proximity in original space).

The authors evaluate in section 5.2.2. the KL divergence of the joint dataset and the do highlight that differences will structurally remain very small because only a small subset of the data is affected by the evaluation. This means the KL divergence over the whole dataset is not the most relevant measure.

Same for section 5.2.3, the assessment metrics should take only OoS points into account.

One limitation present throughout the experiments is the use of very simple datasets with either a low-dimensional sample space (mouse retina) or a simple structure (kNN working out of the box on MNIST). This is very limited to provide empirical evidence that the method represents well OoS data in the created embedding in a more complex setting. Given that this is primarily an empirical paper, this is a major limitation.

The results also seem to indicate that the method is prone to provide embeddings of reduced usefulness. The paper only compares different variants of the methods but not, for instance, a comparison with something like a simple PCA or other more modern techniques for low-dimensional embedding.

Finally, given that the optimization problem solved (equation (4)) is low-dimensional for a small number of OoS points, one could use other techniques to compare quality of baselines, i.e. global optimization solvers would work on these, or multi-start with interior-point methods, which would be pretty similar to a sequential quadratic optimization technique. As an other point, the paper should discuss the relation between the repulsion-free method and sequential quadratic programming, since it essentially seems to consist in that.

**Requested Changes:**

Relate to the existing literature on low-dimensional embedding, compare the developed algorithms with existing counterparts, run empirical evaluations of the method against existing ones, and on datasets that are currently state of the art.

---

> ### Author Response · Authors · 2026-03-29
>
> We thank the Reviewer for evaluating our paper and for their valuable assessment. We will consider their comments and suggestions in order to make the paper more accessible to non experts on the topic of t-SNE. We provide next the response to the raised issues.
>
> **On the precise mathematical meaning:**
>
> We will correct all the raised issues in order to remove any ambiguity. The t-SNE formulation used in the paper is the same as the one used in most papers, including the original one.
> We will also replace “dual” with “complement” when comparing trustworthiness and continuity
>
> **On the use of the KL divergence in Section 5.2.2:**
>
> We agree with the Reviewer that the difference is structurally very small, as it is computed on the whole dataset. However, the difference is still visible and one can compare the different methods in terms of KL divergence, as illustrated in Figures 4, 6c, 7c, 8, 9, and 10.
>
> **On taking only OoS points into account in section 5.2.3:**
>
> We have used the well-known definitions of trustworthiness and continuity, as given in the literature. We prefer not to invent a new definition for these metrics.
> In practice, the obtained results allow to compare all the methods, demonstrating that the used metrics are relevant, as illustrated in Figures 5, 6d, 7d, 8, and 10.
>
> **On the experiments with the use of very simple datasets:**
>
> We have used the MNIST dataset because it is largely used in the literature of t-SNE, thus allowing the community to easily assess the interest of this work and well position it in the associated literature.
> Fashion-MNIST is more complex, as illustrated in its embedding given in Figure 6b, and allows us to provide more insights on the convergence of the algorithms, with some variants of the gradient not reducing the KL divergence.
> We consider the Mouse Retina dataset because it suffers from high class imbalance. It is well known that in t-SNE, the cluster area is proportional to the ratio of the number of training data. Therefore, t-SNE and its OoS extensions could suffer when addressing datasets that have a large imbalance in their classes, which is the case for the Mouse Retina dataset.
>
> If there are more complex settings or datasets proposed by the Reviewer, we will be delighted to include their results in the paper.
>
> **On missing a comparison with a simple PCA or other more modern techniques for low-dimensional embedding:**
>
> The relevance of t-SNE has been demonstrated in many papers, compared to PCA and more modern low-dimensional embedding techniques.
> The contributions of our paper are not on about demonstrating the relevance of t-SNE, but in proposing out-of-sample extensions of t-SNE.
>
> **On the use of other techniques (i.e. global optimization solvers, multi-start with interior-point methods, sequential quadratic optimization technique):**
>
> The optimization techniques proposed by the Reviewer are interesting, but they are out of scope for our paper. As specified in the title of our paper, we propose algorithms ranging from gradient descent to fixed-point iteration.
> Moreover, our optimization problem is not a constrained problem. Therefore, techniques such as interior-point methods and sequential quadratic optimization techniques are not relevant for addressing the out-of-sample problem.
>
> **Requested Changes:**
>
> We have addressed all the raised issues, as described in detail above.

---

### Review · Reviewer_vEpY · 2026-03-10

**Summary Of Contributions:**

The paper studies the problem of computing the t-SNE embedding for a out-of-sample point $x_\ell$ given a dataset $x_1, ..., x_N$ and its t-SNE embeddings $y_1, ..., y_N$.

The paper derives an augmented optimization objective for the purpose and derives a gradient descent algorithm for estimating the embedding for the new sample $x_\ell$. They further show that the gradient can be decomposed into an attraction term and a repulsion term.

The authors claim that the resulting algorithms require a high-degree of hyperparameter fine-tuning (learning rate, momentum etc) to work well. They further claim that these issues can be solved using a fixed-point optimization scheme rather gradient-based optimization. However, the resulting fixed-point equations have no theoretical guarantees (i.e. the fixed-point map is not guaranteed to be a contraction) and the authors also state that they have observed convergence issue.

Moreover, the authors consider a version of the algorithms, where the repulsion-term of the gradient has been left out, and provide a theoretical result showing that the fixed-point algorithm corresponding Newton-like steps on a quadratic surrogate objective. Besides that, there is no theoretical analysis or justification of the proposed method.

Instead, the authors provide a number of numerical experiments based on the three datasets: MNIST, Fashion-MNIST and the Mouse Retina dataset. They provide both qualitative and quantitative results (KL-divergence, continuity and trustworthiness) as well as a couple of robustness/sensitivity experiments.

Strength:
- The paper is well-written, well-structured and easy to follow with a well-defined problem statement.
- The proposed method is simple
- Theorem 1&2 gives some insights into the method.

Weaknesses:
- There is no theoretical analysis or justification of the method and the repulsion is discarded with handwavy arguments
- There are no convergence guarantees, and there authors mention that they observed convergence problems, but this is not described in greater or address in the experiments.

**Additional Comments:**

- Section 3.2: "Since different splits can be carried out, the function f (·) is not unique, leading to different fixed-point iteration algorithms".  It is unclear to me what splits the authors are referring to here.

- Section 5.2.1: "Figure 3f provides the results of the repulsion-free fixed-point iteration after only 5 iterations, illustrating its efficiency to properly embed the OoS images in their respective clusters.". I think it is diffucilt to argue that one embedding is objectively better than the other without more information.

- The legend in Fig. 5 is very small and hard to read

- MNIST and Fashion-MNIST are very similar. It would have been more informative to substitue of the MNIST's for a different dataset.

- A couple of comments regarding the abstract:

	- "We demonstrate the ease of deriving the out-of-sample extension of t-SNE, thanks to the proper nature of t-SNE". It is unclear to me what "the proper nature of t-SNE" refers to.

	- " ... such as demonstrating that its repulsion-free variant corresponds ... to Newton’s method". This seems a little bit misleading to me, since (to the best of my understanding) the repulsion-free variant solves a different problem. So it may correspond to Newton-like updates, but it does so for a difference optimization objective.

**Audience:**

Yes

**Audience Explanation:**

The t-SNE is a very popular algorithm across most areas of ML, and I am convinced the TMLR audience will be interested in new variants of it.

**Claims And Evidence:**

Yes

**Claims Explanation:**

The claims are generally reasonably justified and the approach is generally sound, but I think the convergence aspects and limitations of the method in generally should been described/addressed in greater detail. Moreover, the justification of the discarded the repulsion-term could have been more clear.

The quality of the embeddings is still a little bit unclear to me after reading the paper, but the paper does not really make any claims about the quality of the embedding.

**Requested Changes:**

I think the authors should elaborate on the convergence issues. At least include a discussion of this point, describe your observations and explain/show what happens when the method fails to convergence. How often this it happen?

- The motivation/justification for leaving out the repulsion term is unclear and handwavy (in my opinion). If you need to remove a significant term from the gradient, the cost function seems wrong? Would it have been a more principled approach to change the cost function?  The authors should elaborate on this.

- The repulsion-free algorithm yields a convex combination of the existings embedding. The authors argue that this is a desireable property. But that discussion lacks nuances in my opinion. What if $x_\ell$ is an outlier, then you don't necessarily want to force the point into the convex hull of your existing embedding? The authors should elaborate on this.

- Theorem 1 shows that the repulsion-free fixed point between minimizes a "surrogate" $\mathcal{Q}(y)$. And figure 1 seems to hint that if this process is repeated, then the result will converge to a stationary point of the augmented objective, but can this actually be proven formally?

- Section 5.2.1: "We presume that this is mainly due to the attraction-repulsion terms in the gradient" Since the repulsion-term is very central to this work, I think should be investigated further.

- Section 5.2.2: It is unclear to me why the repulsion-free variant fixed-point variant is so much better compared to the other variants? Also, why have the author not included the fixed point variants of the algorithms containing the repulsion-term for a proper comparison? Same in Fig 6, 7, 8. I would be nice with more details on this.

- For the experiment in Fig 2, it would have been informative to see t-SNE plot computing using the union of the X_train and X_OoS for reference.
- Generally, the paper lacks a discussion of the limitations of the method.

---

> ### Author Response · Authors · 2026-03-29
>
> We thank the Reviewer for evaluating our paper and for their valuable assessment. We will consider their comments and suggestions in order to strengthen the paper, as follows.
>
> ## Requested Changes:
>
> **On the convergence issues:**
>
> We will include in the Final comments (last section) a discussion on the convergence issues and limitations of our method, including convergence analysis in terms of defining a contracting mapping. See comment 2a for Reviewer ZNVw. We did not see any convergence issues in the conducted experiments.
>
> **On the motivation/justification for leaving out the repulsion term:**
>
> We agree with the Reviewer that it would have been a more principled approach to change the cost function, and not to discard the repulsion term as we did.
> However, there are many ML methods that modify the cost function. For example, although variational autoencoders (VAEs) aim to learn the likelihood of the data, this boils downs to two additive terms (the well-known ELBO and KL term). Several researchers have modified it through weighted variants inside the loss, such as $\beta$-VAE for disentangling; As a result, it is no longer the likelihood to be maximized. Other techniques, such as Free bits and KL annealing, also ignore KL when it is below a threshold or during early training stages.
> Moreover, in contrastive learning (and more generally energy-based models), the objective function is decomposed into a positive term (alignment) and a negative term (normalization or partition function). Several methods explicitly discard the latter.
>
> Moreover, our repulsion-free variant can be connected to other methods that do not use repulsion but only attraction:
> - Laplacian Eigenmaps, locally linear embedding… and many other neighborhood-preserving methods consider only the local attraction
> - t-SNE without a repulsion term is studied in several papers, such as (Böhm et al., 2022).
> - While UMAP has both cross-entropy attraction and repulsion terms, there is a weighting hyperparameter denoted negative_sample_rate that can be fine-tuned to balance both, or even set to 0 to remove the repulsion term. The UMAP loss without any negative sampling is studied in papers such as Böhm et al. (2022).
> Reference to be added to our paper:
> Böhm et al., Attraction-Repulsion Spectrum in Neighbor Embeddings, JMLR, 2022.
>
> **On the property of having a convex combination of the existing embedding:**
>
> We agree with the Reviewer that this desirable property could be viewed as a weakness when dealing with outliers. However, one could expect that outliers would be mapped to the frontier of the convex hull, or to some empty space within it. For this reason, we have studied the robustness to out-of-distribution samples in Section 5.4.3. By considering Class 7 as an outlier class, it turns out that the proposed repulsion-free variant provides better results than gradient descent, as shown in Figures 11 and 12.
>
> **On proving formally that the result will converge to a stationary point:**
>
> We were not able to demonstrate this result, although we have explored the optimization through an MM algorithm, which turned out to be harder than expected.
>
> **On investigating further the results in Section 5.2.1:**
>
> Section 5.2.1 provides only qualitative results, which are investigated further in Section 5.2.2 with quantitative results in terms of the KL divergence, and in Section 5.2.3 with quantitative assessment in terms of trustworthiness and continuity.
>
> **On Section 5.2.2:**
>
> The repulsion-free fixed-point variant is so much better compared to the other variants due to several interesting properties given in Section 4, such as the adapted stepsize, as opposed to a fixed one for gradient descent where it is difficult to tune it. Also, gradient descent schemes require an exaggeration.
> In all figures, including Figures 4, 6, 7, and 8, we have considered all the variants of our method, including gradient descent and fixed-point iteration algorithms, grouped in two categories: using both attraction and repulsion terms, and repulsion-free methods. The only missing variant is the fixed–point with both attraction and repulsion terms (17)-(18). As explained on Page 15, it was not included because it suffers from convergence issues, as gradient descent. To enhance convergence, increasing considerably the exaggeration value eventually leads to the repulsion-free.
>
> **On including in Fig. 2 the t-SNE plot computing using the union of the X_train and X_OoS for reference:**
>
> There is no visible difference between the t-SNE computed on 6000 images (currently) and the one computed on 6030 images (proposed by Reviewer).
>
> **On the paper lacking a discussion of the limitations of the method:**
>
> We will include in the Final comments (Section 6) a description of the limitations and weaknesses of the proposed method. See comment 2a for Reviewer ZNVw
>
>
> **Additional Comments:**
>
> We will take into account all comments, not detailed here due to space limitations.

---

### Review · Reviewer_ZNVw · 2026-03-23

**Summary Of Contributions:**

The paper investigates the out-of-sample extension problem for the t-SNE algorithm. Instead of relying on gradient descent optimization, the authors propose a fixed-point iteration method. Furthermore, they introduce a repulsion-free fixed-point algorithm that exhibits more stable convergence and stronger theoretical justification. From a theoretical perspective, the paper shows that the proposed method iteratively minimizes a quadratic objective function. It also establishes connections between the proposed approach, mean shift clustering, and pre-image problems in machine learning. Empirically, the method is evaluated on three different datasets and compared with both gradient descent and repulsion-included algorithms. The experimental results demonstrate the effectiveness of the proposed approach, as well as its fast convergence rate. Additionally, the paper examines the robustness of the method with respect to out-of-domain samples and analyzes its sensitivity to initialization.

**Audience:**

Yes

**Audience Explanation:**

Although proposed decades ago, t-SNE remains one of the most widely used methods for clustering and visualization. Therefore, studying its out-of-sample extension is still of interest to the community, particularly since this paper improves the performance of such extensions.

**Broader Impact Concerns:**

No concerns.

**Claims And Evidence:**

Yes

**Claims Explanation:**

The claim that fixed-point iteration algorithms are better suited for the out-of-sample extension of t-SNE is well supported by the empirical results. The authors compare their methods against multiple baselines across several datasets, demonstrating faster convergence and improved performance. While the overall argument is convincing, there are areas where the paper could be further strengthened:

1. Including comparisons with other state-of-the-art clustering or embedding methods, rather than focusing solely on t-SNE-based approaches, would help better position the contribution of the paper within the broader literature.

2. The current theoretical analysis provides useful insights and connections that help explain how the fixed-point iteration algorithm operates. However, it does not fully justify why the repulsion-free variant achieves better convergence with fewer iterations. A more rigorous theoretical explanation or analysis of this advantage would strengthen the paper.

**Requested Changes:**

1. My laptop became unresponsive when viewing Figures 2, 3, and 7. The authors should consider reducing the size or resolution of these figures to improve readability and accessibility.

2. What are the drawbacks of fixed-point iteration algorithms compared to gradient descent? Additionally, why does the original t-SNE algorithm not use fixed-point iteration for optimization?

---

> ### Author Response · Authors · 2026-03-29
>
> We thank the Reviewer for evaluating our paper and for their valuable assessment. We will consider their comments and suggestions in order to strengthen the paper. We provide next the response to the raised issues.
>
> ## On further strengthening the paper:
>
>  **1. On comparing with clustering methods.**
>
> We agree with the Reviewer that including comparisons with clustering methods would allow us to widen the interest of this work to a larger community of researchers. However, the purpose of our work is not to connect t-SNE to clustering, as this has been largely investigated in the literature, for instance by providing theoretical foundations as demonstrated by Shaham and Steinerberger (2017), by Linderman and Steinerberger (2019), and by Cai and Ma (2022). Although all these studies provide solid foundations showing that t-SNE yields clustering results, t-SNE is not optimized to reveal clusters in data, as shown for instance by Yang, Chen, and Corander (2021). Thus, it is not fair to compare it to clustering techniques.
>
> **2. On justifying the better convergence of the repulsion-free variant.**
>
>   We have provided in the paper several theoretical explanations for why the repulsion-free variant converges better than the attraction-repulsion scheme. More specifically:  In Section 4.1, we show that the repulsion-free variant leads to a solution that belongs to the convex hull of the reference training data, thus constraining the solution so that it does not “overshoot” beyond them. This implicit regularization is not available in the attraction-repulsion scheme. More theoretical results are given and detailed In Section 4.2. All these interesting properties is not verified by the attraction-repulsion scheme.
>
> ## Requested Changes:
>
> **1. On reducing the size or resolution of Figures 2, 3, and 7.**
>
> As requested by the Reviewer, we will reduce the size/resolution of the mentioned figures in order to improve their readability and accessibility.
>
> **2a. On the drawbacks of fixed-point iteration algorithms compared to gradient descent.**
>
>   In general, fixed-point iteration algorithms have the following drawbacks compared to gradient descent:
> - Gradient descent is explicitly aimed at optimization, namely minimizing a cost function. Fixed-point iteration is primarily for solving equations of the form $y = f(y)$ rather than minimizing a cost function. Thus, deriving a fixed-point iteration is generally difficult.
> - Moreover, the construction of the fixed-point iteration $f(y)$ is not unique, and its efficiency heavily depends on its expression, namely on how the problem is reformulated as a fixed-point iteration. To ensure convergence, the function should satisfy certain conditions, such as being a contracting mapping. Gradient descent is also not infallible, but it is easier to operate, with somewhat predictable convergence behavior under suitable learning rates.
> These drawbacks also apply to our method, as discussed in our paper (see for for example the end of Section 3.2). Following the Reviewer’s requested changes, we will include them more explicitly in the final comments of the paper (Section 6).
>
> **2b Additionally, why does the original t-SNE algorithm not use fixed-point iteration for optimization?**
>
> Although the original t-SNE was introduced using a gradient descent scheme (van der Maaten & Hinton, 2008), several variants have been proposed to explore fixed-point iteration for optimization by following different reformulations, such as the HSSNE by Yang et al. (2009), the Laplacian-inspired update rule proposed by van der Maaten (2010), or the non-negative based split of Yang et al. (2010). However, all these methods suffer from the aforementioned drawbacks, which make them less practical. See the top of Page 7 for more details.
> Our repulsion-free method does not suffer from these drawbacks, as demonstrated in the paper through theoretical foundations and experimental results.

---

### Decision · Action_Editor_ua2R · 2026-05-07

**Recommendation:** Accept with minor revision

**Additional Comments:**

Please provide a minor revision will all corrections and adjustments mentioned in the response to the reviewers.

**Audience:**

Yes

**Audience Explanation:**

Yes, as evident by the recommendation of all three reviewers to accept. For instance, as one reviewer mentions, this will be of interest to people interested in data visualization.

**Claims And Evidence:**

Yes

**Claims Explanation:**

Both mathematical derivations are provided as well as extensive numerical results.